# Learning-Augmented Priority Queues

**Ziyad Benomar**
ENSAE, Ecole Polytechnique,
FairPlay joint team
ziyad.benomar@ensae.fr

**Christian Coester**
Department of Computer Science
University of Oxford, UK
christian.coester@cs.ox.ac.uk

## Abstract

Priority queues are one of the most fundamental and widely used data structures in computer science. Their primary objective is to efficiently support the insertion of new elements with assigned priorities and the extraction of the highest priority element. In this study, we investigate the design of priority queues within the learning-augmented framework, where algorithms use potentially inaccurate predictions to enhance their worst-case performance. We examine three prediction models spanning different use cases, and we show how the predictions can be leveraged to enhance the performance of priority queue operations. Moreover, we demonstrate the optimality of our solution and discuss some possible applications.

## 1 Introduction

Priority queues are an essential abstract data type in computer science [Jaiswal, 1968, Brodal, 2013] whose objective is to enable the swift insertion of new elements and access or deletion of the highest priority element. Their applications span a wide range of problems within computer science and beyond. They play a crucial role in sorting [Williams, 1964, Thorup, 2007], in various graph algorithms such as Dijkstra's shortest path algorithm [Chen et al., 2007] or computing minimum spanning trees [Chazelle, 2000], in operating systems for scheduling and load balancing [Sharma et al., 2022], in networking protocols for managing data transmission packets [Moon et al., 2000], in discrete simulations for efficient event processing based on occurrence time [Goh and Thng, 2004], and in implementing hierarchical clustering algorithms [Day and Edelsbrunner, 1984, Olson, 1995].

Various data structures can be used to implement priority queues, each offering distinct advantages and tradeoffs [Brodal, 2013]. However, it is established that a priority queue with $n$ elements cannot guarantee $o(\log n)$ time for all the required operations [Brodal and Okasaki, 1996]. This limitation can be surpassed within the learning augmented framework [Mitzenmacher and Vassilvitskii, 2022], where the algorithms can benefit from machine-learned or expert advice to improve their worst-case performance. We propose in this work learning-augmented implementations of priority queues in three different prediction models, detailed in the next section.

### 1.1 Problem definition

A priority queue is a dynamic data structure where each element $x$ is assigned a key $u$ from a totally ordered universe $(\mathcal{U}, <)$, determining its priority. The standard operations of priority queues are:

   (i) $\mathrm{FindMin}()$: returns the element with the smallest key without removing it,

  (ii) $\mathrm{ExtractMin}()$: removes and returns the element with the smallest key,

 (iii) $\mathrm{Insert}(x, u)$: adds a new element $x$ to the priority queue with key $u$,

 (iv) $\mathrm{DecreaseKey}(x, v)$: decreases the key of an element $x$ to $v$.

38th Conference on Neural Information Processing Systems (NeurIPS 2024).

The elements of the priority queue can be accessed via their keys in $O(1)$ time using a HashMap. Hence, the focus is on establishing efficient algorithms for key storage and organization, facilitating the execution of priority queue operations. For any key $u \in \mathcal{U}$ and subset $\mathcal{V} \subset \mathcal{U}$, we denote by $r(u, \mathcal{V})$ the rank of $u$ in $\mathcal{V}$, defined as the number of keys in $\mathcal{V}$ that are smaller than or equal to $u$,

$$r(u, \mathcal{V}) = \#\{v \in \mathcal{V} : v \leq u\} . \tag{1}$$

The difficulty lies in designing data structures offering adequate tradeoffs between the complexities of the operations listed above. This paper explores how using predictions can allow us to overcome the limitations of traditional priority queues. We examine three types of predictions.

**Dirty comparisons.** In the first prediction model, comparing two keys $(u, v) \in \mathcal{U}^2$ is slow or costly. However, the algorithm can query a prediction of the comparison $(u < v)$. This prediction serves as a rapid or inexpensive, but possibly inaccurate, method of comparing elements of $\mathcal{U}$, termed a *dirty* comparison and denoted by $(u \widehat{<} v)$. Conversely, the true outcome of $(u < v)$ is referred to as a *clean* comparison. For all $u \in \mathcal{U}$ and $\mathcal{V} \subseteq \mathcal{U}$, we denote by $\eta(u, \mathcal{V})$ the number of inaccurate dirty comparisons between $u$ and elements of $\mathcal{V}$,

$$\eta(u, \mathcal{V}) = \#\{v \in \mathcal{V} \ : \ \mathbb{1}(u \widehat{<} v) \neq \mathbb{1}(u < v)\} . \tag{2}$$

This prediction model was introduced in [Bai and Coester, 2023] for sorting, but it has a broader significance in comparison-based problems, such as search [Borgstrom and Kosaraju, 1993, Nowak, 2009, Tschopp et al., 2011], ranking [Wauthier et al., 2013, Shah and Wainwright, 2018, Heckel et al., 2018], and the design of comparison-based ML algorithms [Haghiri et al., 2017, 2018, Ghoshdastidar et al., 2019, Perrot et al., 2020, Meister and Nietert, 2021]. Particularly, it has great theoretical importance for priority queues, which have been extensively studied in the comparison-based framework [Gonnet and Munro, 1986, Brodal and Okasaki, 1996, Edelkamp and Wegener, 2000]. Comparison-based models are often used, for example, when the preferences are determined by human subjects. Assigning numerical scores in these cases is inexact and prone to errors, while pairwise comparisons are absolute and more robust [David, 1963]. Within such a setting, dirty comparisons can be obtained by a binary classifier, and used to minimize human inference yielding clean comparisons, which are time-consuming and might incur additional costs.

**Pointer predictions.** In this second model, upon the addition of a new key $u$ to the priority queue $\mathcal{Q}$, the algorithm receives a prediction $\widehat{\mathrm{Pred}}(u, \mathcal{Q}) \in \mathcal{Q}$ of the predecessor of $u$, which is the largest key belonging to $\mathcal{Q}$ and smaller than $u$. Before $u$ is inserted, the ranks of $u$ and its true predecessor in $\mathcal{Q}$ are equal, hence we define the prediction error as

$$\vec{\eta}(u, \mathcal{Q}) = |r(u, \mathcal{Q}) - r(\widehat{\mathrm{Pred}}(u, \mathcal{Q}), \mathcal{Q})| . \tag{3}$$

In priority queue implementations, a HashMap preserves a pointer from each inserted key to its corresponding position in the priority queue. Consequently, $\widehat{\mathrm{Pred}}(u, \mathcal{Q})$ provides direct access to the predicted predecessor's position. For example, this prediction model finds applications in scenarios where concurrent machines have access to the priority queue [Sundell and Tsigas, 2005, Shavit and Lotan, 2000, Lindén and Jonsson, 2013]. Each machine can estimate, within the elements it has previously inserted, which one precedes the next element it intends to insert. However, this estimation might not be accurate, as other concurrent machines may have inserted additional elements.

**Rank predictions.** The last setting assumes that the priority queue is used in a process where a finite number $N$ of distinct keys will be inserted, i.e., the priority queue is not used indefinitely, but $N$ is unknown to the algorithm. Upon the insertion of any new key $u_i$, the algorithm receives a prediction $\widehat{R}(u_i)$ of the rank of $u_i$ among all the $N$ keys $\{u_j\}_{j \in [N]}$. Denoting by $R(u_i) = r(u_i, \{u_j\}_{j \in [N]})$ the true rank, the prediction error of $u_i$ is

$$\eta^{\Delta}(u_i) = |R(u_i) - \widehat{R}(u_i)| . \tag{4}$$

The same prediction model was explored in [McCauley et al., 2024] for the online list labeling problem. Bai and Coester [2023] investigate a similar model for the sorting problem, but in an offline setting where the $N$ elements to sort and the predictions are accessible to the algorithm from the start.

Note that $N$ counts the distinct keys added with Insert or DecreaseKey operations. An arbitrarily large number of DecreaseKey operations thus can make $N$ arbitrarily large although the total number

of insertions is reduced. However, with a lazy implementation of DecreaseKey, we can assume without loss of generality that $N$ is at most quadratic in the total number of insertions. Indeed, it is possible to omit executing the DecreaseKey operations when queried, and only store the new elements' keys to update. Then, at the first ExtractMin operation, all the element's keys are updated by executing for each one only the last queried DecreaseKey operation involving it. Denoting by $k$ the total number of insertions, there are at most $k$ ExtractMin operations, and for each of them, there are at most $k$ DecreaseKey operations executed. The total number of effectively executed DecreaseKey operations is therefore $O(k^2)$. In particular, this implies that a complexity of $O(\log N)$ is also logarithmic in the total number of insertions.

## 1.2 Our results

We first investigate augmenting binary heaps with predictions. To leverage dirty comparisons, we first design a *randomized binary search* algorithm to find the position of an element $u$ in a sorted list $L$. We prove that it terminates using $O(\log |L|)$ dirty comparisons and $O(\log \eta(u, L))$ clean comparisons in expectation. Subsequently, we use this result to establish an insertion algorithm in binary heaps using $O(\log \log n)$ dirty comparisons and reducing the number of clean comparisons to $O(\log \log \eta(u, \mathcal{Q}))$. However, ExtractMin still mandates $O(\log n)$ clean comparisons. In the two other prediction models, binary heaps and other heap implementations of priority queues appear unsuitable, as the positions of the keys are not determined solely by their ranks.

Consequently, in Section 3, we shift to using skip lists. We devise randomized insertion algorithms requiring, in expectation, only $O(\log \vec{\eta}(u, \mathcal{Q}))$ time and comparisons in the pointer prediction model, $O(\log n)$ time and $O(\log \eta(u, \mathcal{Q}))$ clean comparisons in the dirty comparison model, and $O(\log \log N + \log \max_{i \in [N]} \eta^\Delta(u_i))$ time and $O(\log \max_{i \in [N]} \eta^\Delta(u_i))$ comparisons in the rank prediction model, where we use in the latter an auxiliary van Emde Boas (vEB) tree [van Emde Boas et al., 1976]. Across the three prediction models, FindMin and ExtractMin only necessitate $O(1)$ time, and the complexity of DecreaseKey aligns with that of insertion. Finally, we prove in Theorem 3.4 the optimality of our data structure. Table 1.2 summarizes the complexities of our learning-augmented priority queue (LAPQ) in the three prediction models compared to standard priority queue implementations. The complexity of FindMin is $O(1)$ for all the listed priority queues.

| Priority queues | ExtractMin | Insert | DecreaseKey |
|:---:|:---:|:---:|:---:|
| Binary Heap | $O(\log n)$ | $O(\log \log n)$ | $O(\log n)$ |
| Fibonacci Heap (amortized) | $O(\log n)$ | $O(1)$ | |
| Skip List (average) | $O(1)$ | $O(\log n)$ | |
| LAPQ with dirty comparisons (average) | $O(1)$ | $O(\log \eta(u, \mathcal{Q}))$ | |
| LAPQ with pointer predictions (average) | $O(1)$ | $O(\log \vec{\eta}(u, \mathcal{Q}))$ | |
| LAPQ with rank predictions (average) | $O(1)$ | $O(\log \max_{i \in [n]} \eta^\Delta(u_i))$ | |

Table 1: Number of comparisons per operation used by different priority queues.

Our learning-augmented data structure enables additional operations beyond those of priority queues, such as the *maximum priority queue* operations FindMax, ExtractMax, and IncreaseKey with analogous complexities, and removing an arbitrary key $u$ from the priority queue, finding its predecessor or successor in expected $O(1)$ time.

Furthermore, we show in Section 4.1 that it can be used for sorting, yielding the same guarantees as the learning-augmented sorting algorithms presented in [Bai and Coester, 2023] for the positional predictions model with *displacement error*, and for the dirty comparison model. In the second model, our priority queue offers even stronger guarantees, as it maintains the elements sorted at any time even if the insertion order is adversarial, while the algorithm of [Bai and Coester, 2023] requires a random insertion order to achieve a sorted list by the end within the complexity guarantees. We also show how the learning-augmented priority queue can be used to accelerate Dijkstra's algorithm.

Finally, in Section 5, we compare the performance of our priority queue using predictions with binary and Fibonacci heaps when used for sorting and for Dijkstra's algorithm on both real-world city maps and synthetic graphs. The experimental results confirm our theoretical findings, showing that adequately using predictions significantly reduces the complexity of priority queue operations.

## 1.3 Related work

In this section, we briefly discuss related works on learning-augmented algorithms and priority queues. For a more extensive review of related work, please refer to Appendix A.

**Learning-augmented algorithms.** Learning-augmented algorithms, introduced in [Lykouris and Vassilvtiskii, 2018, Purohit et al., 2018], have captured increasing interest over the last years, as they allow breaking longstanding limitations in many algorithm design problems. Assuming that the decision-maker is provided with potentially incorrect predictions regarding unknown parameters of the problem, learning-augmented algorithms must be capable of leveraging these predictions if they are accurate (consistency), while keeping the worst-case performance without advice even if the predictions are arbitrarily bad or adversarial (robustness). While many fundamental online problems were studied in this setting (see Appendix A), the design of data structures with predictions remains relatively underexplored. The seminal paper by [Kraska et al., 2018] shows how predictions can be used to optimize space usage. Another study by [Lin et al., 2022] demonstrates that the runtime of binary search trees can be optimized by incorporating predictions of item access frequency. Recent papers have extended this prediction model to other data structures, such as dictionaries [Zeynali et al., 2024] and skip lists [Fu et al., 2024]. The prediction models we study in the current paper deviate from the latter, and are more related to those considered respectively in [Bai and Coester, 2023] for sorting, and [McCauley et al., 2024] for online list labeling. An overview of the growing body of work on learning-augmented algorithms (also known as algorithms with predictions) is maintained at [Lindermayr and Megow, 2022].

**Priority queues implementations.** Binary heaps, introduced by Williams [1964], are one of the first efficient implementations of priority queues. They allow all the operations in $O(\log n)$ time, where $n$ is the number of items in the queue. A first improvement was introduced with Binomial heaps [Vuillemin, 1978], reducing the amortized time of insertion to $O(1)$. A breakthrough came later with Fibonacci heaps [Fredman and Tarjan, 1987], which allow all the operations in constant amortized time, except for ExtractMin, which takes $O(\log n)$ time. However, Fibonacci heaps are known to be slow in practice [Larkin et al., 2014], and other implementations with weaker theoretical guarantees such as binary heaps are often preferred. Another possible implementation uses *skip lists* [Pugh, 1990], which are probabilistic data structures, guaranteeing in expectation a constant time for FindMin and ExtractMin, and $O(\log n)$ time for Insert and DecreaseKey.

## 2 Heap priority queues

A common implementation of priority queues uses binary heaps, enabling all operations in $O(\log n)$ time. Binary heaps maintain a balanced binary tree structure, where all depth levels are fully filled, except possibly for the last one, to which we refer in all this section as the *leaf level*. Moreover, it satisfies the *heap property*, i.e., any key is smaller than all its children. To maintain these two structure properties, when a new element is added, it is first inserted in the leftmost empty position in the leaf level, and then repeatedly swapped with its parent until the heap property is restored.

### 2.1 Insertion in the comparison-based model

Insertion in a binary heap can be accomplished using only $O(\log \log n)$ comparisons, albeit $O(\log n)$ time, by doing a binary search of the new element's position along the path from the leftmost empty position in the leaf level to the root, which is a sorted list of size $O(\log n)$. To improve the insertion complexity with dirty comparisons, we first tackle the search problem in this setting.

**Search with dirty comparisons.** Consider a sorted list $L = (v_1, \ldots, v_k)$ and a target $u$, the position of $u$ in $L$ can be found using binary search with $O(\log k)$ comparisons. Extending ideas from Bai and Coester [2023] and Lykouris and Vassilvtiskii [2018], this complexity can be reduced using dirty comparisons. Indeed, we can obtain an estimated position $\widehat{r}(u, L)$ through a binary search with dirty comparisons, followed by an exponential search with clean comparisons, starting from $\widehat{r}(u, L)$ to find the exact position $r(u, L)$. However, the positions of inaccurate dirty comparisons can be adversarially chosen to compromise the algorithm. This can be addressed by introducing randomness to the dirty search phase. We refer to the *randomized binary search* as the algorithm that proceeds

similarly to the binary search, but whenever the search is reduced to an array $\{v_i, \ldots, v_j\}$, instead of comparing $u$ to the pivot $v_m$ with index $m = i + \lfloor \frac{j-i}{2} \rfloor$, it compares $u$ to a pivot with an index chosen uniformly at random in the range $\{i + \lceil \frac{j-i}{4} \rceil, \ldots, j - \lceil \frac{j-i}{4} \rceil\}$.

**Theorem 2.1.** *A dirty randomized binary search followed by a clean exponential search finds the target's position using $O(\log k)$ dirty comparisons and $O(\log \eta(u, L))$ clean comparisons in expectation.*

**Randomized insertion in a binary heap.** In a binary heap $\mathcal{Q}$, any new element $u$ is always inserted along the path of size $O(\log n)$ from the root to the leftmost empty position in the leaf level. If all the inaccurate dirty comparisons are chosen along this path, then the insertion would require $O(\log \eta(u, \mathcal{Q}))$ clean comparisons by Theorem 2.1. This complexity can be reduced further by randomizing the choice of the root-leaf path where $u$ is inserted, as explained in Algorithm 1.

---

**Algorithm 1:** Randomized insertion in binary heap

---

**Input:** Binary heap $\mathcal{Q}$ with randomly filled positions in the leaf level, new key $u$

1   $L \leftarrow$ keys path from a uniformly random empty position in the leaf level to the root;
2   $\widehat{r}(u, L) \leftarrow$ outcome of a **dirty randomized** binary search of $u$ in $L$;
3   $r(u, L) \leftarrow$ outcome of the **clean** exponential search of $u$ in $L$ starting from index $\widehat{r}(u, L)$;
4   Insert $u$ in the chosen leaf empty position, then swap it with its parents until position $r(u, L)$;

---

**Theorem 2.2.** *The insertion algorithm 1 requires $O(\log n)$ time, $O(\log \log n)$ dirty comparisons and $O(\log \log \eta(u, \mathcal{Q}))$ clean comparisons in expectation.*

### 2.2 Limitations

The previous theorem demonstrates that accurate predictions can reduce the number of clean comparisons for insertion in a binary heap. However, for the $\mathrm{ExtractMin}$ operation, when the minimum key, which is the root of the tree, is deleted, its two children are compared and the smallest is placed in the root position, and this process repeats recursively, with each new empty position filled by comparing both of its children, requiring necessarily $O(\log n)$ clean comparisons in total to ensure the heap priority remains intact. Improving the efficiency of $\mathrm{ExtractMin}$ using dirty comparisons would therefore require bringing major modifications to the binary heap's structure.

Similar difficulties arise when attempting to enhance $\mathrm{ExtractMin}$ using dirty comparisons or the other prediction models in different heap implementations, such as Binomial or Fibonacci heaps. Consequently, we explore in the next section another priority queue implementation, using skip lists, which allows for an easier and more efficient exploitation of the predictions.

## 3 Skip lists

A priority queue can be implemented naively by maintaining a dynamic sorted linked list of keys. This guarantees constant time for $\mathrm{ExtractMin}$, but $O(n)$ time for insertion. Skip lists (see Figure 1) offer a solution to this inefficiency, by maintaining multiple levels of linked lists, with higher levels containing fewer elements and acting as shortcuts to lower levels, facilitating faster search and insertion in expected $O(\log n)$ time. In all subsequent discussions concerning linked lists or skip lists, it is assumed that they are doubly linked, having both predecessor and successor pointers between elements.

The first level in a skip list is an ordinary linked list containing all the elements, which we denote by $v_1, \ldots, v_n$. Every higher level is constructed by including each element from the previous level independently with probability $p$, typically set to $1/2$. For any key $v_i$ in the skip list, we define its height $h(v_i)$ as the number of levels where it appears, which is an independent geometric random variable with parameter $p$. A number $2h(v_i)$ of pointers are associated with $v_i$, giving access to the previous and next element in each level $\ell \in [h(v_i)]$, denoted respectively by $\mathrm{Prev}(v_i, \ell)$ and $\mathrm{Next}(v_i, \ell)$. Using a HashMap, these pointers can be accessed in $O(1)$ time via the key value $v_i$. For convenience, we consider that the skip list contains two additional keys $v_0 = -\infty$ and $v_{n+1} = \infty$, corresponding respectively to the head and the NIL value. Both have a height equal to the maximum height in the queue $h(v_0) = h(v_{n+1}) = \max_{i \in [n]} h(v_i)$.

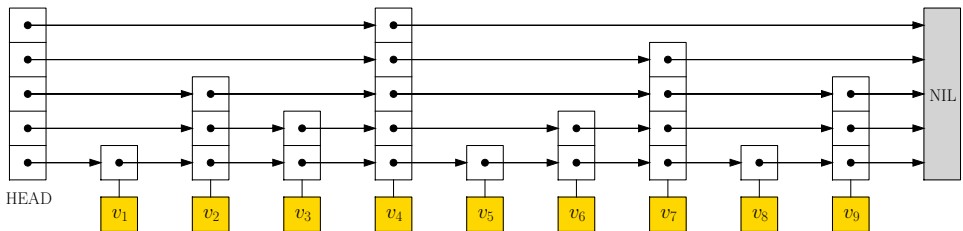

Figure 1: A skip list with keys $v_1 < \ldots < v_9 \in \mathcal{U}$.

Since the expected height of keys in the skip list is $1/p$, deleting any key only requires a constant time in expectation, by updating its associated pointers, along with those of its predecessors and successors in the levels where it appears. In particular, FindMin and ExtractMin take $O(1)$ time, and DecreaseKey can be performed by deleting the element and reinserting it with the new key, yielding the same complexity as insertion. Furthermore, by the same arguments, inserting a new key $u$ next to a given key $v_i$ in the skip list can be done in expected constant time.

Therefore, implementing efficient Insert and DecreaseKey operations for skip lists with predictions is reduced to designing efficient search algorithms to find the predecessor of a target key $u$ in the skip list, i.e., the largest key $v_i$ in the skip list that is smaller than $u$. In all the following, we denote by $\mathcal{Q}$ a skip list containing $n$ keys $v_1 \leq \ldots \leq v_n \in \mathcal{U}$, and $u \in \mathcal{U}$ the target key. As explained in Appendix C.1, we can assume without loss of generality that the keys $(u, v_1, \ldots, v_n)$ are pairwise distinct. In the following, we present separately for each model an insertion algorithm leveraging the predictions.

### 3.1 Pointer prediction

Given a pointer prediction $v_j = \widehat{\text{Pred}}(u, \mathcal{Q})$, we describe below an algorithm for finding the true predecessor of $u$ starting from the position of the key $v_j$, then inserting $u$. We assume in the algorithm that $v_j \leq u$. If $v_j > u$, then the algorithm can be easily adapted by reversing the search direction.

---

**Algorithm 2:** ExpSearchInsertion$(\mathcal{Q}, v_j, u)$

**Input:** Skip list $\mathcal{Q}$, source $v_j \in \mathcal{Q}$, and new key $u \in \mathcal{U}$

1   $w \leftarrow v_j$;                                           ▷ Bottom-Up search
2   **while** $\text{Next}(w, h(w)) \leq u$ **do**
3     $\mid$   $w \leftarrow \text{Next}(w, h(w))$;
4   $\ell \leftarrow h(w)$;                                   ▷ Top-Down search
5   **while** $\ell > 0$ **do**
6     $\mid$   **while** $\text{Next}(w, \ell) \leq u$ **do**
7     $\mid$     $\mid$   $w \leftarrow \text{Next}(w, \ell)$;
8     $\mid$   $\ell \leftarrow \ell - 1$;
9   Insert $u$ next to $w$;

---

Algorithm 2 is inspired by the classical exponential search in arrays. The first phase consists of a bottom-up search, expanding the size of the search interval by moving to upper levels until finding a key $w \in \mathcal{Q}$ satisfying $w \leq u < \text{Next}(w, h(w))$. The second phase conducts a top-down search from level $h(w)$ downward, refining the search until locating the position of $u$. It is worth noting that the classical search algorithm in skip lists, denoted by $\text{Search}(\mathcal{Q}, u)$, corresponds precisely to the top-down search, starting from the head of the skip list instead of $w$.

**Theorem 3.1.** *Augmented with pointer predictions, a skip list allows* FindMin *and* ExtractMin *in expected $O(1)$ time, and* Insert$(u)$ *in expected $O(\log \vec{\eta}(u, \mathcal{Q}))$ time using Algorithm 2.*

## 3.2 Dirty comparisons

We devise in this section a search algorithm using dirty and clean comparisons. Algorithm 3 first estimates the position of $u$ with a dirty top-down search starting from the head, then performs a clean exponential search starting from the estimated position to find the true position.

---

**Algorithm 3:** Insertion with dirty and clean comparisons

---

**Input:** Skip list $\mathcal{Q}$, new key $u \in \mathcal{U}$

**1** $\hat{w} \leftarrow \mathrm{Search}(\mathcal{Q}, u)$ with dirty comparisons;

**2** $\mathrm{ExpSearchInsertion}(\mathcal{Q}, \hat{w}, u)$ with clean comparisons;

---

The dirty search concludes within $O(\log n)$ steps, and Theorem 3.1 guarantees that the exponential search terminates within $O(\log |r(\hat{w}, \mathcal{Q}) - r(u, \mathcal{Q})|)$ steps. Combining these results and relating the distance between $u$ and $\hat{w}$ in $\mathcal{Q}$ to the prediction error $\eta(u, \mathcal{Q})$, we derive the following theorem.

**Theorem 3.2.** *Augmented with dirty comparisons, a skip list allows* $\mathrm{FindMin}$ *and* $\mathrm{ExtractMin}$ *in* $O(1)$ *expected time, and* $\mathrm{Insert}(u)$ *with Algorithm 3 in* $O(\log n)$ *expected time, using* $O(\log n)$ *dirty comparisons and* $O(\log \eta(u, \mathcal{Q}))$ *clean comparisons in expectation.*

## 3.3 Rank predictions

In the rank prediction model, each $\mathrm{Insert}(u)$ request is accompanied by a prediction $\widehat{R}(u)$ of the rank of $u$ among all the distinct keys already in, or to be inserted into the priority queue. If the predictions are accurate and the total number $N$ of distinct keys to be inserted is known, the problem reduces to designing a priority queue with integer keys in $[N]$, taking as keys the ranks $(R_i)_{i \in [N]}$. This problem can be addressed using a van Emde Boas (vEB) tree over $[N]$ [van Emde Boas et al., 1976], which requires $O(\log \log N)$ time for insertion, deletion, finding the minimum or maximum, and finding the predecessor or successor of any element, guaranteeing in particular $O(\log \log N)$ time for all priority queue operations. More details on its structure can be found in Appendix C.5.

To leverage rank predictions, we use an auxiliary vEB tree $\mathcal{T}$ along with the skip list $\mathcal{Q}$. All the insertion and deletion operations are made simultaneously on $\mathcal{T}$ and $\mathcal{Q}$. However, the priorities used in $\mathcal{T}$ are the predicted ranks $\{\widehat{R}(u_i)\}_{i \in [N]}$. Whenever a new key $u_i$ is to be added, Algorithm 4 inserts it first in $\mathcal{T}$ at position $\widehat{R}(u_i)$, gets its predecessor $\hat{w}$ in $\mathcal{T}$, i.e., the element in $\mathcal{T}$ with the largest predicted rank smaller than or equal to $\widehat{R}(u_i)$, then uses $\hat{w}$ as a pointer prediction to find the position of $u_i$ in $\mathcal{Q}$. If the predecessor is not unique, the algorithm chooses an arbitrary one.

---

**Algorithm 4:** Insertion with rank prediction

---

**Input:** Skip list $\mathcal{Q}$, vEB tree $\mathcal{T}$ on $[N]$, new element $u_i \in \mathcal{U}$, prediction $\widehat{R}(u_i) \in [N]$

**1** Insert $u_i$ in $\mathcal{T}$ with key $\widehat{R}(u_i)$;

**2** $\hat{w} \leftarrow$ predecessor of $u_i$ in $\mathcal{T}$;

**3** $\mathrm{ExpSearchInsertion}(\mathcal{Q}, \hat{w}, u_i)$;

---

We explain in Appendix C.5 how the data structure can be adapted when $N$ is unknown and the rank predictions are not necessarily in $[N]$, and we prove the following theorem, giving both the runtime and comparison complexities of the priority queue operations using this data structure.

**Theorem 3.3.** *If* $\widehat{R}(u_i) = O(N)$ *for all* $i \in [N]$*, then there is a data structure allowing* $\mathrm{FindMin}$ *and* $\mathrm{ExtractMin}$ *in* $O(1)$ *amortized time, and* $\mathrm{Insert}$ *in* $O(\log \log N + \log \max_{i \in [N]} \eta^{\Delta}(u_i))$ *amortized time using* $O(\log \max_{i \in [N]} \eta^{\Delta}(u_i))$ *comparisons in expectation.*

In contrast to other prediction models, the complexity of inserting $u_i$ is not impacted only by $\eta^{\Delta}(u_i)$, but by the maximum error over all keys $\{u_j\}_{j \in [N]}$. This occurs because the exponential search conducted in Algorithm 4 starts from the key $\hat{w} \in \mathcal{Q}$, whose error also affects insertion performance. A similar behavior is observed in the online list labeling problem [McCauley et al., 2024], where the bounds provided by the authors also depend on the maximum prediction error for insertion.

With perfect predictions, the number of comparisons for insertion becomes constant, and its runtime $O(\log \log N)$. It is not clear if the runtime of all the priority queue operations can be reduced to $O(1)$ with perfect predictions. Indeed, the problem in that case is reduced to a priority queue with all the keys in $[N]$. The best-known solution to this problem is a randomized priority queue, by Thorup [2007], supporting all operations in $O(\sqrt{\log \log N})$ time. However, in our approach, we use vEB trees beyond the classical priority queue operations, as we also require fast access to the predecessor of any element. A data structure supporting all these operations solves the dynamic predecessor problem, for which vEB trees are optimal [Pătraşcu and Thorup, 2006]. Reducing the runtime of insertion below $O(\log \log N)$ would therefore require omitting the use of predecessor queries.

### 3.4 Lower bounds

As explained earlier, ExtractMin requires only $O(1)$ expected time in skip lists. Furthermore, we presented insertion algorithms for the three prediction models and provided upper bounds on their complexities. The following theorem establishes lower bounds on the complexities of ExtractMin and Insert for any priority queue augmented with any of the three prediction types.

**Theorem 3.4.** *For each of the three prediction models, the following lower bounds hold.*

(i) *Dirty comparisons: no data structure $\mathcal{Q}$ can support ExtractMin with $O(1)$ clean comparisons and Insert$(u)$ with $o(\log \eta(u, \mathcal{Q}))$ clean comparisons in expectation.*

(ii) *Pointer predictions: no data structure $\mathcal{Q}$ can support ExtractMin with $O(1)$ comparisons and Insert$(u)$ with $o(\log \vec{\eta}(u, \mathcal{Q}))$ comparisons in expectation.*

(iii) *Rank predictions: no data structure $\mathcal{Q}$ can support ExtractMin with $O(1)$ comparisons and Insert$(u_i)$ with $o(\log \max_{i \in [N]} \eta^{\Delta}(u_i))$ comparisons in expectation, for all $i \in [N]$.*

These lower bounds with Theorems 3.2 and 3.1 prove the tightness of our priority queue in the dirty comparison and the pointer prediction models. In the rank prediction model, the comparison complexities proved in Theorem 3.3 are optimal, whereas the runtimes are only optimal up to an additional $O(\log \log N)$ term. In particular, they are optimal if the maximal error is at least $\Omega(\log N)$.

## 4 Applications

### 4.1 Sorting algorithm

Our learning-augmented priority queue can be used for sorting a sequence $A = (a_1, \ldots, a_n)$, by first inserting all the elements, then repeatedly extracting the minimum until the priority queue is empty. We compare below the performance of this sorting algorithm to those of [Bai and Coester, 2023].

**Dirty comparison model.** Denoting by $\eta_i = \eta(a_i, A)$, Bai and Coester [2023] prove a sorting algorithm using $O(n \log n)$ time, $O(n \log n)$ dirty comparisons, and $O(\sum_i \log(\eta_i + 2))$ clean comparisons. Theorem 3.2 yields the same guarantees with our learning-augmented priority queue. Moreover, our learning-augmented priority queue is a skip list, maintaining elements in sorted order at any time, even if the elements are revealed online and the insertion order is adversarial, while in [Bai and Coester, 2023], it is crucial that the insertion order is chosen uniformly at random.

**Positional predictions.** In their second prediction model, they assume that the algorithm is given offline access to predictions $\{\widehat{R}(a_i)\}_{i \in [n]}$ of the relative ranks $\{R(a_i)\}_{i \in [n]}$ of the $n$ elements to sort, and they study two different error measures. The rank prediction error $\eta_i^{\Delta} = |R(a_i) - \widehat{R}(a_i)|$ matches their definition of *displacement error*, for which they prove a sorting algorithm in $O(\sum_i \log(\eta_i^{\Delta} + 2))$ time. The same bound can be deduced using our results in the pointer prediction model. Further discussion on this claim can be found in Appendix E.

**Online rank predictions.** If $n$ is unknown to the algorithm, and the elements $a_1, \ldots, a_n$ along with their predicted ranks are revealed online, possibly in an adversarial order, then by Theorem 3.3, the total runtime of our priority queue for maintaining all the inserted elements sorted at any time is $O(n \log \log n + n \log \max_i \eta_i^{\Delta})$, and the number of comparisons used is $O(n \log \max_i \eta_i^{\Delta})$. No analogous result is demonstrated in [Bai and Coester, 2023] in this setting.

## 4.2 Dijkstra's shortest path algorithm

Consider a run of Dijkstra's algorithm on a directed positively weighted graph $G$ with $n$ nodes and $m$ edges. The elements inserted into the priority queue are the nodes of the graph, and the corresponding keys are their tentative distances to the source, which are updated over time. During the algorithm's execution, at most $m + 1$ distinct keys $\{d_i\}_{i \in [m+1]}$ are inserted into the priority queue. Given online predictions $(\widehat{R}(d_i))_{i \in [m+1]}$ of their relative ranks $(R(d_i))_{i \in [m+1]}$, the total runtime using our priority queue augmented with rank predictions is

$$O\big(m \log \log n + m \log \max_{i \in [m+1]} |R(d_i) - \widehat{R}(d_i)|\big) .$$

In contrast, the shortest path algorithm of Lattanzi et al. [2023] (which also works for negative edges) has a linear dependence on a similar error measure. Even with arbitrary error, our guarantee is never worse than the $O(m \log n)$ runtime with binary heaps. Using Fibonacci heaps results in an $O(n \log n + m)$ runtime, which is surpassed by our learning-augmented priority queue in the case of sparse graphs where $m = o(\frac{n \log n}{\log \log n})$ if predictions are of high quality. However, it is known that Fibonacci heaps perform poorly in practice, even compared to binary heaps, as supported by our experiments.

## 5 Experiments

In this section, we empirically evaluate the performance of our learning-augmented priority queue (LAPQ) by comparing it with Binary and Fibonacci heaps. We use two standard benchmarks for this evaluation: sorting and Dijkstra's algorithm. For the sorting benchmark, we also compare our results with those from Bai and Coester [2023]. For Dijkstra's algorithm, we assess performance on both real city maps and synthetic random graphs. In all the experiments, each data point represents the average result from 30 independent runs. Additional experiments and a detailed discussion on the prediction models and the obtained results can be found in Appendix F. The code used for conducting the experiments is available at github.com/Ziyad-Benomar/Learning-augmented-priority-queues.

**Sorting.** We compare sorting using our LAPQ with the algorithms of Bai and Coester [2023] under their same experimental settings. Given a sequence $A = (a_1, \ldots, a_n)$, we evaluate the complexity of sorting it with predictions in the *class* and the *decay* setting. In the first, $A$ is divided into $c$ classes $((t_{k-1}, t_k])_{k \in [c]}$, where $0 = t_0 \leq t_1 \leq \ldots \leq t_c = n$ are uniformly random thresholds. The predicted rank of any item $a_i$ with $t_k \leq i < t_{k+1}$ is sampled uniformly at random within $(t_k, t_{k+1}]$. In the decay setting, the ranking is initially accurate but degrades over time. Each time step, one item's predicted position is perturbed by 1, either left or right, uniformly at random.

In both settings, we test the LAPQ with the three prediction models. First, assuming that the rank predictions are given offline, we use pointer predictions as explained in Appendix E. In the second case, the elements to insert along with their predicted ranks are revealed online in a uniformly random order. Finally, we test the dirty comparison setting with the dirty order $(a_i \mathbin{\widehat{<}} a_j) = (\widehat{R}(a_i) < \widehat{R}(a_j))$.

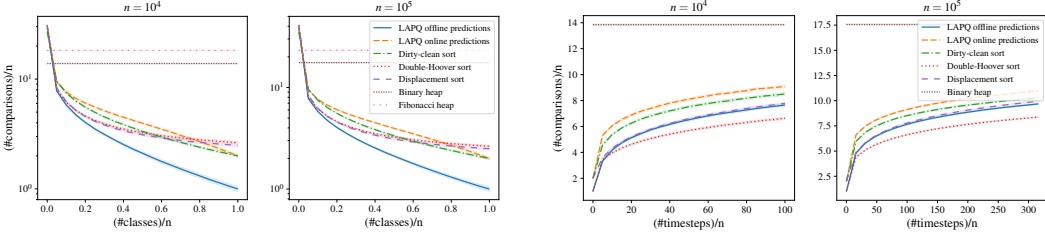

Figure 2: Sorting in the *class* setting     Figure 3: Sorting in the *decay* setting

Figures 2 and 3 show the obtained results respectively in the class and the decay setting for $n \in \{10^4, 10^5\}$. In the class setting with offline predictions, the LAPQ slightly outperforms the *Double-Hoover* and *Displacement* sort algorithms of Bai and Coester [2023], which were shown to outperform classical sorting algorithms. In the decay setting, the LAPQ matches the performance

of the Displacement sort, but is slightly outperformed by the Double-Hoover sort. With online predictions, although the problem is harder, LAPQ's performance remains comparable to the previous algorithms. In both settings, the LAPQ with offline predictions, online predictions, and dirty comparisons all yield better performance than binary or Fibonacci heaps, even with predictions that are not highly accurate.

**Dijkstra's algorithm.** Consider a graph $G = (V, E)$ with $n$ nodes and $m$ edges, and a source node $s \in V$. In the first predictions setting, we pick a random node $\hat{s}$ and run Dijkstra's algorithm with $\hat{s}$ as the source, memorizing all the keys $\hat{D} = (\hat{d}_1, \ldots, \hat{d}_m)$ inserted into the priority queue. In subsequent runs of the algorithm with different sources, when a key $d_i$ is to be inserted, we augment the insertion with the rank prediction $\widehat{R}(d_i) = r(d_i, \hat{D})$. We call these *key rank predictions*. This model aims at exploiting the topology and uniformity of city maps. As computing shortest paths from any source necessitates traversing all graph edges, keys inserted into the priority queue—partial sums of edge lengths—are likely to exhibit some degree of similarity even if the algorithm is executed from different sources. Notably, this prediction model offers an explicit method for computing predictions, readily applicable in real-world scenarios.

In the second setting, we consider rank predictions of the nodes in $G$, ordered by their distances to $s$. As Dijkstra's algorithm explores a new node $x \in V$, it receives a prediction $\widehat{r}(x)$ of its rank. The node $x$ is then inserted with a key $d_i$, to which we assign the prediction $\widehat{R}(d_i) = \widehat{r}(x)$. Unlike the previous experimental settings, we initially have predictions of the nodes' ranks, which we extend to predictions of the keys' ranks. Similarly to the sorting experiments, we consider *class* and *decay* perturbations of the node ranks.

In the context of searching the shortest path, rank predictions in the class setting can be derived from subdividing the city into multiple smaller areas. Each class corresponds to a specific area, facilitating the ordering of areas from closest to furthest relative to the source. However, comparing the distances from the source to the nodes in the same class might be inaccurate. On the other hand, the decay setting simulates modifications to shortest paths, such as rural works or new route constructions, by adding or removing edges from the graph. These alterations may affect the ranks of a limited number of nodes, which corresponds to the time steps in the decay setting.

We present below the experiment results obtained with the maps of Paris and London. More experiments with additional city maps and synthetic graphs are in Appendix F. The city maps were obtained using the Python library Osmnx [Boeing, 2017]. Figures 4 and 5 respectively illustrate the results in the *class* and the *decay* settings with *node rank predictions*. In both figures, for each city, we present the numbers of comparisons used for the same task by a binary and Fibonacci heap, and the number of comparisons used when the priority queue is augmented with *key rank predictions*.

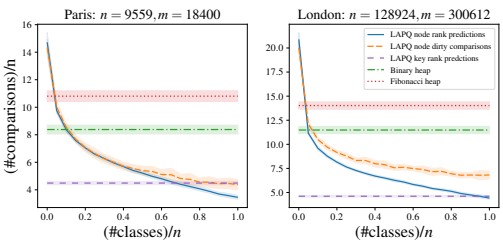

Figure 4: Shortest path, *class* predictions

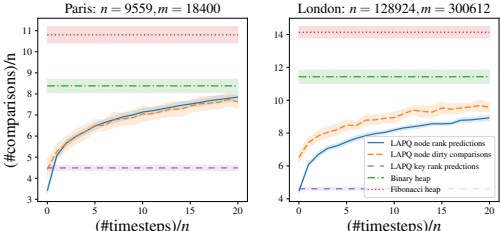

Figure 5: Shortest path, *decay* predictions

In both settings, the performance of the LAPQ substantially improves with the quality of the predictions, and notably, *key rank predictions* yield almost the same performance as perfect *node rank predictions*, affirming our intuition on the similarity between the keys inserted in runs of Dijkstra's algorithm starting from different sources.

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

# Learning-Augmented Priority Queues

## Appendix

## A Extended related work

**Learning-augmented algorithms** Learning-augmented algorithms, introduced in [Lykouris and Vassilvtiskii, 2018, Purohit et al., 2018], have captured increasing interest over the last years, as they allow breaking longstanding limitations in many algorithm design problems. Assuming that the decision-maker is provided with potentially incorrect predictions regarding unknown parameters of the problem, learning-augmented algorithms must be capable of leveraging these predictions if they are accurate (consistency), while keeping the worst-case performance without advice even if the predictions are arbitrarily bad or adversarial (robustness). Many fundamental online problems were studied in this setting, such as ski rental [Gollapudi and Panigrahi, 2019, Anand et al., 2020, Bamas et al., 2020, Diakonikolas et al., 2021, Antoniadis et al., 2021, Maghakian et al., 2023, Shin et al., 2023], scheduling [Purohit et al., 2018, Merlis et al., 2023, Lassota et al., 2023, Benomar and Perchet, 2024b], matching [Antoniadis et al., 2020, Dinitz et al., 2021, Chen et al., 2022], and caching [Lykouris and Vassilvtiskii, 2018, Chlkedowski et al., 2021, Bansal et al., 2022, Antoniadis et al., 2023b,a, Christianson et al., 2023]. Data structures can also be improved within this framework. However, this remains underexplored compared to online algorithms. The seminal paper by Kraska et al. [2018] shows how predictions can be used to optimize space usage. Another study by [Lin et al., 2022] demonstrates that the runtime of binary search trees can be enhanced by incorporating predictions of item access frequency. Recent papers have extended this prediction model to other data structures, such as dictionaries [Zeynali et al., 2024] and skip lists [Fu et al., 2024]. The prediction models we study deviate from the latter, and are more related to those considered respectively in [Bai and Coester, 2023] for sorting, and [McCauley et al., 2024] for online list labeling. An overview of the growing body of work on learning-augmented algorithms (also known as algorithms with predictions) is maintained at [Lindermayr and Megow, 2022].

**Dirty comparisons** The dirty and clean comparison model is also related to the prediction querying model, gaining a growing interest in the study of learning-augmented algorithms, where the decision-maker decides when to query predictions and for which items [Im et al., 2022, Benomar and Perchet, 2024a, Sadek and Elias, 2024]. In particular, having free-weak and costly-strong oracles has been studied in [Silwal et al., 2023] for correlation clustering, in [Bateni et al., 2023] for clustering and computing minimum spanning trees in a metric space, and in [Eberle et al., 2024] for matroid optimization. Another related setting involves the algorithm observing partial information online, which it can then use to decide whether to query a costly hint about the current item. This has been explored in contexts such as online linear optimization [Bhaskara et al., 2021] and the multicolor secretary problem [Benomar et al., 2023].

**Priority queues** Binary heaps, introduced by Williams [1964], are one of the first efficient implementations of priority queues. They allow $\mathrm{FindMin}$ in constant time, and the other operations in $O(\log n)$ time, where $n$ is the number of items in the queue. Shortly after, other heap-based implementations with similar guarantees were designed, such as leftist heaps [Crane, 1972] and randomized meldable priority queues [Gambin and Malinowski, 1998]. A new idea was introduced in [Vuillemin, 1978] with Binomial heaps, where instead of having a single tree storing the items, a binomial heap is a collection of $\Theta(\log n)$ trees with exponentially growing sizes, all satisfying the heap property. They allow insertion in constant amortized time, and $O(\log n)$ time for $\mathrm{ExtractMin}$ and $\mathrm{DecreaseKey}$. A breakthrough came with Fibonacci heaps [Fredman and Tarjan, 1987], which allow all the operations in constant amortized time, except for $\mathrm{ExtractMin}$, which takes $O(\log n)$ time. It was shown later, in works such as [Brodal, 1996, Brodal et al., 2012], that the same guarantees can be achieved in the worst-case, not only in amortized time. Although they offer very good theoretical guarantees, Fibonacci heaps are known to be slow in practice [Larkin et al., 2014, Lewis,

2023], and other implementations with weaker theoretical guarantees such as binary heaps are often preferred. We refer the interested reader to the detailed survey on priority queues by Brodal [2013].

**Skip lists** A skip list [Pugh, 1990] is a probabilistic data structure, based on classical linked lists, having shortcut pointers allowing fast access between non-adjacent elements. Due to the simplicity of their implementation and their strong performance, skip lists have many applications [Hu et al., 2003, Ge and Zdonik, 2008, Basin et al., 2017]. In particular, they can be used to implement priority queues [Rönngren and Ayani, 1997], guaranteeing in expectation a constant time for FindMin and ExtractMin, and $O(\log n)$ time for Insert and DecreaseKey. They show particularly good performance compared to other implementations in the case of concurrent priority queues, where multiple users or machines can make requests to the priority queue [Shavit and Lotan, 2000, Lindén and Jonsson, 2013, Zhang and Dechev, 2015].

# B  Heap priority queues

## B.1  Exponential search and randomized binary search

The insertion algorithm we propose for binary heaps combines the classical exponential search and our randomized adaptation of the binary search algorithm. We provide a brief overview of the exponential search and its runtime, followed by an analysis of the runtime upper bound for our randomized binary search algorithm.

**Exponential search.** The exponential search of a target $u$ in a sorted list $L = (v_1, \ldots, v_k)$ starting from index $i \in [k]$ consists in first comparing $u$ to $v_i$, and if $v_i < u$ then $u$ is compared to $v_{i+1}, v_{i+2}, v_{i+4}, \ldots$ until an integer $\ell$ is found satisfying $v_{i+2^\ell} < u \le v_{i+2^{\ell+1}}$, a binary search is then conducted between indices $i + 2^\ell$ and $i + 2^{\ell+1}$ to find the position of $u$. If the first comparison shows instead that $u < v_i$, then $u$ is compared to $v_{i-1}, v_{i-2}, v_{i-4}, \ldots$. Denoting by $j$ the true position of $u$ in $L$, the exponential search terminates in $O(\log|i - j|)$ time.

**Randomized binary search (RBS).** Consider a sorted list $L = (v_1, \ldots, v_k) \in \mathcal{U}^k$ and a target $u \in \mathcal{U}$. When the search of $u$ in $L$ is reduced to the sub-list $(v_i, \ldots, v_j)$, RBS samples an index

$$\ell \sim \mathcal{U}\text{niform}\left(\{i + \lceil\tfrac{j-i}{4}\rceil, \ldots, j - \lceil\tfrac{j-i}{4}\rceil\}\right) ,$$

if $u = v_\ell$ then algorithm terminates, if $u < v_\ell$ then the search is reduced to the sub-list $(v_i, \ldots, v_{\ell-1})$, otherwise it is reduced to $(v_{\ell+1}, \ldots, v_j)$. This process iterates until the obtained sub-list has a size of 1, the position of $u$ is then immediately deduced by comparing it to the single element in the sub-list.

**Lemma B.1.** *RBS in a list of size $k$ terminates in $O(\log k)$ time with probability $1$.*

*Proof.* Let us denote by $S_t$ the size of the sub-list to which the search is reduced after $t$ steps, and $T = \min\{t \ge 1 : S_t = 1\}$. For convenience, we consider that $S_t = 1$ for all $t > T$. It holds that $S_0 = n$, and for all $t \ge 1$, if $S_t = j - i + 1 \ge 2$ then

$$S_{t+1} \le \max(j - \ell, \ell - i) \le j - i - \lceil\tfrac{j-i}{4}\rceil \le \tfrac{3}{4}(j - i) \le \tfrac{3}{4}S_t .$$

On the other hand, if $S_t = 1$ then $S_{t+1} = 1$. We deduce that $S_{t+1} \le \max(1, \tfrac{3}{4}S_t)$ with probability 1, hence $S_t \le \max(1, (3/4)^t k)$ for all $t \ge 1$. Therefore, it holds almost surely that

$$T \le \lceil\log_{4/3} k\rceil .$$

$\square$

## B.2  Proof of Theorem 2.1

*Proof.* The algorithm described in Theorem 2.1 runs RBS with dirty comparisons to obtain an estimated position $\widehat{r}(u, L)$ of $u$ in $L$, then it conducts a clean exponential search starting from $\widehat{r}(u, L)$ to find the exact position $r(u, L)$. Lemma B.1 guarantees that the dirty RBS uses $O(\log k)$ comparisons, while the clean exponential search terminates in expectation after $O(\log|r(u, L) - \widehat{r}(u, L)|)$ comparisons. Therefore, for demonstrating Theorem 2.1, it suffices to prove that $\mathbb{E}[\log|r(u, L) - \widehat{r}(u, L)|] = O(\log \eta(u, L))$.

To lighten the expressions, we write $\eta$ instead of $\eta(u, L)$ in the rest of the proof. Let $(i_0, j_0) = (1, n)$, and $(i_t, j_t)$ the indices delimiting the sub-list of $L$ to which the RBS is reduced after $t$ steps for all $t \geq 1$. Denoting by $t^* + 1$ the first step where a dirty comparison is inaccurate, it holds that $\widehat{r}(u, L), r(u, L) \in \{i_{t^*}, j_{t^*}\}$. Indeed, the first $t^*$ dirty comparisons are all accurate, thus the estimated and the true positions of $u$ in $L$ are both in $\{i_{t^*}, j_{t^*}\}$. Therefore, $|r(u, L) - \widehat{r}(u, L)| \leq S_{t^*} - 1$, where $S_t = j_t - i_t + 1$ is the size of the sub-list delimited by indices $(i_t, j_t)$, which yields

$$\log |r(u, L) - \widehat{r}(u, L)| \leq \log(S_{t^*}) . \tag{5}$$

We focus in the remainder on bounding $\mathbb{E}[\log(S_{t^*})]$. For this, we use a proof scheme similar to that of Lemmas A.4 and A.5 in Bai and Coester [2023].

$$
\begin{aligned}
\mathbb{E}[\log(S_{t^*})] &\leq \mathbb{E}[\lceil \log(S_{t^*}) \rceil] \\
&\leq \sum_{m \geq 1}^{\infty} m \Pr(\lceil \log(S_{t^*}) \rceil = m) \\
&\leq \log(\eta + 1) + \sum_{m = \lceil \log(\eta+1) \rceil}^{\infty} m \Pr(\lceil \log(S_{t^*}) \rceil = m) ,
\end{aligned} \tag{6}
$$

and for all $m \geq 1$, we have that

$$
\begin{aligned}
\Pr(\lceil \log(S_{t^*}) \rceil = m) &= \Pr(S_{t^*} \in (2^{m-1}, 2^m]) \\
&= \sum_{t=1}^{\infty} \Pr(S_t \in (2^{m-1}, 2^m], t = t^*) \\
&= \sum_{t=1}^{\infty} \Pr(S_t \in (2^{m-1}, 2^m]) \Pr(t = t^* \mid S_t \in (2^{m-1}, 2^m]) .
\end{aligned}
$$

In the sum above, for any $t \geq 1$, the probability that $t = t^*$ is the probability that the next sampled pivot index $\ell_t$ is in the set $\mathcal{F}(u, L) = \{v \in L : \mathbb{1}(u \stackrel{\sim}{<} v) \neq \mathbb{1}(u < v)\}$, which has a cardinal $\eta$. Thus

$$\Pr(t = t^* \mid S_t) = \Pr(\ell_t \in \mathcal{F}(u, L) \mid S_t) = \frac{\#(\mathcal{F} \cap \{i_t, \ldots, j_t\})}{S_t} \leq \frac{\eta}{S_t} .$$

It follows that

$$
\begin{aligned}
\Pr(\lceil \log(S_{t^*}) \rceil = m) &\leq \frac{\eta}{2^{m-1}} \sum_{t=1}^{\infty} \Pr(S_t \in (2^{m-1}, 2^m]) \\
&\leq \frac{\eta}{2^{m-1}} \mathbb{E}[\#\{t \geq 1 : S_t \in (2^{m-1}, 2^m]\}] \\
&\leq \frac{3\eta}{2^{m-1}} ,
\end{aligned}
$$

where the last inequality is an immediate consequence of the identity $S_{t+1} \leq \frac{3}{4} S_{t+1}$ proved in Lemma B.1, which shows in particular that $S_{t+3} < S_t/2$, i.e. the after at most 3 steps, the size of the search sub-list is divided by two, hence $(S_t)_t$ falls into the interval $(2^{m-1}, 2^m]$ at most three times. Substituting into (6) gives

$$
\begin{aligned}
\mathbb{E}[\log S_{t^*}] &\leq \log(\eta + 1) + 3\eta \cdot \sum_{m = \lceil \log(\eta+1) \rceil}^{\infty} \frac{m}{2^{m-1}} \\
&= \log(\eta + 1) + 3\eta \cdot O\left(\frac{\log \eta}{\eta}\right) \\
&= O(\log \eta) .
\end{aligned}
$$

Therefore, by (5), the expected number of comparisons used during the clean exponential search is at most $O(\mathbb{E}[\log |r(u, L) - \widehat{r}(u, L)|]) = O(\log \eta)$, which concludes the proof. $\square$

## B.3 Proof of Theorem 2.2

*Proof.* Consider a binary heap $\mathcal{Q}$ containing $n$ elements, with positions randomly filled in the leaf level. Even with this randomization, $\mathcal{Q}$ is still balanced, and the length of any path from an empty position in the leaf level to the root has a size $O(\log n)$. Denoting by $L$ the random insertion path chosen in Algorithm 1, constructing $L$ and storing it requires $O(\log n)$ time and space. By Theorem 2.1, the randomized search in Algorithm 1 uses $O(\log \log n)$ dirty comparisons, and the exponential search uses an expected number of $O(\mathbb{E}[\log \eta(u, L)])$ clean comparisons. Inserting $u$ and then swapping it up until its position does not use any comparison and requires a $O(\log n)$ time. Therefore, to prove the theorem, it suffices to demonstrate that $\mathbb{E}[\log \eta(u, L)] = O(\log \log \eta(u, \mathcal{Q}))$.

We assume that the key $u$ to be inserted can be chosen by an oblivious adversary, unaware of the randomization outcome. This means that the internal state of $\mathcal{Q}$ remains private at any time. Let $\mathcal{F}(u, \mathcal{Q}) = \{v \in \mathcal{Q} : \mathbb{1}(u \stackrel{\frown}{<} v) \neq \mathbb{1}(u < v)\}$, which has a cardinal of $\eta(u, \mathcal{Q})$. Enumerating the binary heap levels starting from the root, each level $\ell$ except for the last one, denoted $\ell_{\max}$, contains exactly $2^{\ell-1}$ elements $v_1^\ell, \dots, v_{2^{\ell-1}}^\ell$. A key $v_i^\ell$ in level $\ell$ has a probability $1/2^{\ell-1}$ of belonging to $L$. Denoting by $\xi_i^\ell = \mathbb{1}(v_i^\ell \in \mathcal{F}(u, \mathcal{Q}))$, the expected number of keys in $L$ belonging to $\mathcal{F}(u, \mathcal{Q})$ is

$$\mathbb{E}[\eta(u, L)] = \sum_{\ell=1}^{\ell_{\max}} \sum_{i=1}^{2^{\ell-1}} \frac{\xi_i^\ell}{2^{\ell-1}} = \sum_{\ell=1}^{\ell_{\max}} \frac{1}{2^{\ell-1}} \left( \sum_{i=1}^{2^{\ell-1}} \xi_i^\ell \right) .$$

Given that $\sum_{\ell=1}^{\ell_{\max}} \sum_{i=1}^{2^{\ell-1}} \xi_i^\ell = \eta(u, \mathcal{Q}) \leq 2^{\lceil \log(\eta(u, \mathcal{Q})+1) \rceil} - 1$, the expression above is maximized under this constraint for the instance $\bar{\xi}_i^\ell = \mathbb{1}(\ell \leq \lceil \log(\eta(u, \mathcal{Q})+1) \rceil)$. Therefore,

$$\mathbb{E}[\eta(u, L)] \leq \sum_\ell \frac{1}{2^{\ell-1}} \left( \sum_{i=1}^{2^{\ell-1}} \bar{\xi}_i^\ell \right) = \lceil \log(\eta(u, \mathcal{Q})+1) \rceil .$$

Finally, Jensen's inequality and the concavity of log yield

$$\mathbb{E}[\log \eta(u, L)] \leq \log \mathbb{E}[\eta(u, L)] = O(\log \log \eta(u, \mathcal{Q})) ,$$

which gives the result.

$\square$

# C Skip lists

## C.1 Unique keys in the priority queue

The keys of the priority queue can be considered pairwise distinct by grouping elements with the same key together in a collection, for example, a HashSet. This collection can be accessible in $O(1)$ time via the key using a HashMap. When a new item $x$ with priority $u$ is to be inserted, the algorithms first checks if the key $u$ is already in the priority queue, if that is the case then $x$ is simply added to the collection corresponding to $u$ in $O(1)$ time. Otherwise, the key $u$ must first be inserted into its correct position.

With such implementation, when an ExtractMin operation is called, if multiple elements correspond to the minimum key, then the algorithm can for example retrieve an arbitrary one of them, of the first inserted one, depending on the use case.

## C.2 Expected maximum of i.i.d. geometric random variables

Before presenting the insertion algorithms in the three prediction models, we present an upper bound from [Eisenberg, 2008] on the expected maximum of i.i.d. geometric random variables with parameter $p$. This Lemma will be useful in the analysis of our algorithms, as the heights of elements in the skip list are i.i.d. geometric random variables. The following Lemma is an immediate consequence of

**Lemma C.1** ([Eisenberg, 2008]). *If $X_1, \dots, X_m$ are i.i.d. random variables following a geometric random distribution with parameter $p$, then, denoting by $q = 1 - p$, it holds that*

$$\mathbb{E}[\max_{i \in [m]} X_i] \leq 1 + \frac{1}{\log(1/q)} \sum_{k=1}^{m} \frac{1}{k} = O(\log_{1/q} m) .$$

## C.3 Pointer prediction model

*Proof of Theorem 3.1.* The key $w$ found at the end of the algorithm is the processor of $u$ in $\mathcal{Q}$. Inserting $u$ next to $w$ only requires expected $O(1)$ time. Thus, we demonstrate in the following that Algorithm 2, starting from a key $v_j \in \mathcal{Q}$, finds the predecessor of $u$ in $O(\log |r(v_j) - r(u)|)$ expected time, where $r(v)$ denotes the rank $r(v, \mathcal{Q})$ of $v$ in $\mathcal{Q}$. In particular, for $v_j = \widehat{\mathrm{Pred}}(u, \mathcal{Q})$, we obtain the claim of the theorem.

We assume in the proof that $v_j < u$, i.e. the exponential search goes from left to right. Let $h^*(v_j, u)$ be the maximum height of all elements in $\mathcal{Q}$ between $u$ and $v_j$

$$h^*(v_j, u) = \max\{h(v) : v \in \mathcal{Q} \text{ such that } v_j \leq v \leq u\} .$$

The number of elements between $v_j$ and $u$, with $v_j$ included, is $|r(u) - r(v_j)| + 1$, and the heights of all the elements in $\mathcal{Q}$ are independent geometric random variables with parameter $p$, thus Lemma C.1 gives that $\mathbb{E}[h^*(v_j, u)] = O(\log |r(u) - r(v_j)|)$.

The key $w^*$ found at the end of the Bottom-Up search is the last element, going from $v_j$ to $u$, having a height of $h^*(v_j, u)$. Indeed, in the Bottom-Up search, whenever the algorithm reaches a new key, it moves to the maximum level to which the key belongs, the height of $w^*$ is therefore necessarily the maximum height of all the keys between $v_j$ and $w^*$, i.e. $h(w^*) = h^*(v_j, w^*)$. Since the Bottom-Up search stops at key $w^*$, then $\mathrm{Next}(w^*, h(w^*)) > u$, which means that there is no key in $\mathcal{Q}$ between $w^*$ and $u$ having a height more than $h(w^*) - 1$.

The number of comparisons made in this phase is therefore at most the number of comparisons needed to reach level $h^*(v_j, u) + 1$ starting from $v_j$ using the Bottom-Up search. We consider the hypothetical setting where the skip list is infinite to the right, the expected number of comparisons to reach level $h^*(v_j, u) + 1$, in this case, is an upper bound on the expected number of comparisons needed in the Bottom-Up phase of Algorithm 2, as the algorithm also terminates if the end of the skip-list is reached. Let $T(\ell)$ be the expected number of comparisons made in the bottom-up search to reach level $\ell$ in an infinite skip list. After each comparison made in the bottom-up search, it is possible to go at least one level up with probability $p$, while the algorithm can only move horizontally to the right with probability $1 - p$. This induces the inequality

$$T(\ell) \leq 1 + pT(\ell - 1) + (1 - p)T(\ell) ,$$

which yields

$$T(\ell) \leq \frac{1}{p} + T(\ell - 1) .$$

Given that $T(1) = 0$, we have for $\ell \geq 1$ that $T(\ell) \leq \frac{\ell - 1}{p}$, and we deduce that the expected number of comparisons made by the algorithm during the Bottom-Up search is at most $\frac{\mathbb{E}[h^*(v_j, u)]}{p} = O(\log |r(v_j) - r(u)|)$.

In the Top-Down search described in the second phase, the path traversed by the algorithm is exactly the inverse of the Bottom-Up search from the predecessor of $u$ to $w^*$. The same arguments as the analysis of the first phase give that the Top-Down search terminates after $O(\log |r(v_j) - r(u)|)$ comparisons in expectation, which concludes the proof. □

## C.4 Dirty comparison model

*Proof of Theorem 3.2.* Let $\mathcal{F}(u, \mathcal{Q})$ the set of keys in $\mathcal{Q}$ whose dirty comparisons with $u$ are inaccurate

$$\mathcal{F}(u, \mathcal{Q}) = \{v \in \mathcal{Q} : \mathbb{1}(u \mathbin{\widehat{<}} v) \neq \mathbb{1}(u < v)\} .$$

The prediction error $\eta(u, \mathcal{Q})$ defined in (2) is the cardinal of $\mathcal{F}(u, \mathcal{Q})$. Let

$$h^*(\mathcal{F}(u, \mathcal{Q})) = \max\{h(v) : v \in \mathcal{F}(u, \mathcal{Q})\}$$

be the maximal height of elements in $\mathcal{F}(u, \mathcal{Q})$. The search algorithm $\mathrm{Search}(\mathcal{Q}, u)$ with dirty comparisons starts from the highest level at the head of the skip list, and then goes down the different levels until finding the predicted position $\hat{w}$ of $u$. This is the classical search algorithm in skip lists, and it is known to require $O(\log n)$ comparisons to terminate. This can also be deduced from the

analysis of the exponential search described in Algorithm 2, as it corresponds to the Top-Down search starting from the head of the skip list.

Before level $h^*(\mathcal{F}(u, \mathcal{Q}))$ is reached in $\mathrm{Search}(\mathcal{Q}, u)$, all the dirty comparisons are accurate. Denoting by $v'$ the last key in $\mathcal{Q}$ visited in a level higher than $h^*(\mathcal{F}(u, \mathcal{Q}))$ during this search, and $v'' = \mathrm{Next}(\mathcal{Q}, v', h^*(\mathcal{F}(u, \mathcal{Q})))$, it holds that both keys $\hat{w}$ and $w$ are between $v'$ and $v''$, and there is no key in $\mathcal{Q}$ between $v'$ and $v''$ with height more than $h^*(\mathcal{F}(u, \mathcal{Q})) - 1$.

In particular, the maximal key height between $\hat{w}$ and $w$ is at most $h^*(\mathcal{F}(u, \mathcal{Q})) - 1$. We showed in the proof of Theorem 3.1 that the number of comparisons and runtime of $\mathrm{ExpSearchInsertion}(\mathcal{Q}, v_j, u)$ is linear with the maximal height of keys in $\mathcal{Q}$ that are between $v_j$ and $u$. Using this result with $\hat{w}$ instead of $v_j$ gives that $\mathrm{ExpSearchInsertion}(\mathcal{Q}, \hat{w}, u)$ finds the position of $u$ using $O(h^*(\mathcal{F}(u, \mathcal{Q})))$ clean comparisons. Finally, since $h^*(\mathcal{F}(u, \mathcal{Q}))$ is the maximum of a number $\eta(u, \mathcal{Q})$ of i.i.d. geometric random variables with parameter $p$, Lemma C.1 gives that

$$\mathbb{E}[h^*(\mathcal{F}(u, \mathcal{Q}))] = O(\log \eta(u, \mathcal{Q})) \ ,$$

which proves the theorem. $\qquad\square$

### C.5 Rank prediction model

To leverage rank predictions, as explained in Section 3.3, we use an auxiliary van Emde Boas (vEB) tree van Emde Boas [1977]. We describe in the following the structure ad complexity of vEB trees on $[N]$, and we explain how they can be adapted in the case where $N$ is unknown.

#### C.5.1 Van Emde Boas (vEB) trees

A vEB tree over an interval $\{i + 1, \ldots, i + m\}$ has a root with $\sqrt{m}$ children, each being the root of a smaller vEB tree over a sub-interval $\{i + k\sqrt{m} + 1, \ldots, i + (k+1)\sqrt{m}\}$ for some $k \in \{0, \ldots, \sqrt{m} - 1\}$. The tree leaves are either empty or contain elements with the corresponding key, stored together in a collection, and internal nodes carry binary information indicating whether or not the subtree they root contains at least one element. Denoting by $H(m)$ the height of a vEB tree of size $m$, it holds that $H(m) = 1 + H(\sqrt{m})$, which yields that $H(m) = O(\log \log m)$, enabling efficient implementation of the operations listed below in $O(\log \log m)$ time:

- $\mathrm{Insert}(x, k)$: insert a new element $x$ with key $k \in [m]$ in the tree,
- $\mathrm{Delete}(x, k)$: Delete the element/key pair $(x, k)$,
- $\mathrm{Predecessor}(k)$: return the element in the tree with the largest key smaller than or equal to $k$,
- $\mathrm{Successor}(k)$: return the element in the tree with the smallest key larger than or equal to $k$,
- $\mathrm{ExtractMin}()$: removes and returns the element with the smallest key.

Other operations such as $\mathrm{FindMin}$, $\mathrm{FindMax}$, or $\mathrm{Lookup}(k)$ are supported in $O(1)$ time. These runtimes, however, require knowing the maximal key value $m$ from the beginning, as it is used for constructing $\mathcal{T}_m$.

#### C.5.2 Dynamic size vEB trees

If the maximal key value $\bar{R}$ is unknown, we argue that the operations listed above can be supported in amortized $O(\log \log \bar{R})$ time. Given a vEB tree $\mathcal{T}_m$ of size $m$, if a new key $k \in \{m + 1, \ldots, 2m\}$ is to be inserted, we construct an empty vEB tree $\mathcal{T}_{2m}$ of size $2m$ in $O(m)$ time, then repeatedly extract the elements with minimal key from $\mathcal{T}_m$ and insert them in $\mathcal{T}_{2m}$ with the same key. Each $\mathrm{ExtractMin}$ operation in $\mathcal{T}_m$ and insertion in $\mathcal{T}_{2m}$ requires $O(\log \log m)$ time. Therefore, constructing $\mathcal{T}_{2m}$ and inserting all the elements from $\mathcal{T}_m$ takes $O(m \log \log m)$ time.

This observation can be used to define a vEB with dynamic size. First, we construct a vEB tree with an initial constant size $R_0$. If at some point the size of the vEB tree is $m \geq R_0$ and a new key $k > m$ is to be inserted, then we iterate the size doubling process described before until the size of the vEB tree is at least $k$. At any time step, denoting by $\bar{R}$ the maximal key value inserted in the vEB tree, and letting $i \geq 1$ such that $2^{i-1} R_0 \leq \bar{R} < 2^i R_0$, the size of the tree has been doubled up to this step $i$

times to cover all the keys. The total time for resizing the vEB tree is at most proportional to

$$\sum_{j=0}^{i-1} 2^j R_0 \log \log (2^j R_0) \le \Big( \sum_{j=0}^{i-1} 2^j \Big) R_0 \log \log \bar{R}$$

$$\le 2^i R_0 \log \log \bar{R}$$

$$\le 2\bar{R} \log \log \bar{R} \ .$$

Therefore, if $N$ is the total number of elements inserted into the vEB tree and $\bar{R}$ the maximum key value, we can neglect the cost of resizing by considering that each insertion requires an amortized time of $O((1 + \frac{\bar{R}}{N}) \log \log \bar{R})$. The runtime of all the other operations is $O(\log \log \bar{R})$. In particular, if $\bar{R} = O(N)$, then all the operations run in $O(\log \log N)$ amortized time.

### C.5.3 Priority queue with rank predictions

Consider the setting where $N$ is unknown and all the predicted ranks, revealed online, satisfy $R(u_i) = O(N)$. The priority queue we consider is a skip list $\mathcal{Q}$ with an auxiliary dynamic vEB tree $\mathcal{T}$. For insertions, we use Algorithm 4. For ExtractMin, we first extract the minimum $u_{\min}$ from $\mathcal{Q}$ in $O(1)$ time, then we delete it from the corresponding position $\widehat{R}(u_{\min})$ in $\mathcal{T}$ in $O(\log \log N)$ time. Deleting an arbitrary key $u$ from $\mathcal{Q}$ can be done in the same way, by removing it from $\mathcal{Q}$ in expected $O(1)$ time and then deleting it from the position $\widehat{R}(u)$ in $\mathcal{T}$ in $O(\log \log N)$ time. Thus, as in the other prediction models, DecreaseKey can be implemented by deleting the element and reinserting it with the new key, which requires the same complexity as insertion, with an additional $O(\log \log N)$ term.

We will prove that the priority queue described above yields the claim of Theorem 3.3. We assume that the keys $u_1, \dots, u_N$ are inserted in this order. For all $t \in [N]$, we denote by $\mathcal{Q}^t, \mathcal{T}^t$ the set of keys in the skip list and the set of integer keys in the dynamic vEB tree right after the insertion of $u_t$, with the keys in $\mathcal{Q}^t$ or $\mathcal{T}^t$. Note that, due to eventual deletions, the sizes of $\mathcal{Q}^t$ and $\mathcal{T}^t$ can be smaller than $t$.

Following the definition of the rank in (1), for all $i, t \in [N]$, we denote by $r(u_i, \mathcal{Q}^t)$ the rank of $u_i$ in $\mathcal{Q}^t$, and $r(\widehat{R}(u_i), \mathcal{T}^t)$ the rank of $\widehat{R}(u_i)$ in $\mathcal{T}^t$. The following lemma shows that the absolute difference between the two previous quantities for any given $i, t$ is at most twice the maximal rank prediction error.

**Lemma C.2.** *For any subset $I \subset [N]$, it holds for all $i \in I$ that*

$$|r(u_i, \{u_j\}_{j \in I}) - r(\widehat{R}(u_i), \{\widehat{R}(u_j)\}_{j \in I})| \le 2 \max_{j \in [N]} \eta^\Delta(u_j) \ .$$

*Proof.* Let

$$\Delta_* = \max_{I \subset [N]} \Big( \max_{i \in I} |r(u_i, \{u_j\}_{j \in I}) - r(\widehat{R}(u_i), \{\widehat{R}(u_j)\}_{j \in I})| \Big) \ , \tag{7}$$

and let $I \subset [N]$ for which this maximum is reached. We assume without loss of generality that $I = [m]$. For all $s \in [N]$, let

$$\tilde{\mathcal{Q}}^s = \{u_j\}_{j \in [s]}, \quad \tilde{\mathcal{T}}^s = \{\widehat{R}(u_j)\}_{j \in [s]}, \quad \Delta^s = \max_{i \in [s]} |r(u_i, \tilde{\mathcal{Q}}^s) - r(\widehat{R}(u_i), \tilde{\mathcal{T}}^s)| \ .$$

To simplify the expressions, we denote by $r_i^s = r(u_i, \tilde{\mathcal{Q}}^s)$ and $\widehat{r}_i^s = r(\widehat{R}(u_i), \tilde{\mathcal{T}}^s)$ for all $(s, i) \in [N]^2$. We will prove that $\Delta^s = \Delta_*$ for all $s \in \{m, \dots, N\}$. By definition of $\Delta_*$, it holds that $\Delta_* \ge \Delta^s$ for all $s$, it remains to prove the other inequality. It is true for $s = m$ by definition of $I$ and $m$. Now let $s \in \{m, \dots, N-1\}$ and assume that $\Delta^s = \Delta_*$, i.e. there exists $i \le s$ such that $|r_i^s - \widehat{r}_i^s| = \Delta_*$. Assume for example that $\widehat{r}_i^s = r_i^s + \Delta_*$.

- If $u_{s+1} < u_i$, then $r_{s+1}^{s+1} < r_i^{s+1}$ and $r_i^{s+1} = r_i^s + 1$. By definition of $\Delta_*$, it holds that

$$\widehat{r}_{s+1}^{s+1} \le r_{s+1}^{s+1} + \Delta_* \le r_i^{s+1} - 1 + \Delta_* = r_i^s + \Delta_* = \widehat{r}_i^s \ .$$

This implies necessarily that $\widehat{R}(u_{s+1}) \le \widehat{R}(u_i)$, and therefore

$$\widehat{r}_i^{s+1} = \widehat{r}_i^s + 1 = r_i^s + 1 + \Delta_* = r_i^{s+1} + \Delta_* ,$$

which gives that $\Delta^{s+1} \ge |\widehat{r}_i^{s+1} - r_i^{s+1}| = \Delta_*$.

- If $u_{s+1} > u_i$, then $r_i^{s+1} = r_i^s$ and $\widehat{r}_i^{s+1} \ge \widehat{r}_i^s$, thus $\Delta^{s+1} \ge \widehat{r}_i^{s+1} - r_i^{s+1} \ge \widehat{r}_i^s - r_i^s = \Delta_*$.

- If $u_{s+1} = u_i$ then $r_{s+1}^{s+1} = r_i^{s+1} = r_i^s + 1$. On the other hand, if $\widehat{R}(u_{s+1}) \le \widehat{R}(u_i)$ then $\widehat{r}_i^{s+1} = \widehat{r}_i^s + 1$, otherwise $\widehat{r}_{s+1}^{s+1} \ge \widehat{r}_i^s + 1$. In both cases, it holds that

$$\begin{aligned}
\Delta^{s+1} &\ge \max(\widehat{r}_i^{s+1} - r_i^{s+1}, \, \widehat{r}_{s+1}^{s+1} - r_{s+1}^{s+1}) \\
&\ge (\widehat{r}_i^s + 1) - (r_i^s + 1) = \Delta_* .
\end{aligned}$$

The same proof can be used for the case where $r_i^s = \widehat{r}_i^s + \Delta_*$. Therefore, we have for all $s \in \{m, \ldots, N\}$ that $\Delta_* = \Delta^s$. In particular, for $s = N$, observing that $r(u_i, \tilde{\mathcal{Q}}^N) = R(u_i)$ for all $i \in [N]$, we obtain

$$\Delta_* = \max_{i \in [N]} |R(u_i) - r(\widehat{R}(u_i), \tilde{\mathcal{T}}^N)| . \tag{8}$$

Let us denote by $\eta_{\max}^\Delta = \max_{j \in [N]} \eta^\Delta(u_j)$ the maximum rank prediction error. We will prove in the following that

$$\forall i \in [N]: \quad |R(u_i) - r(\widehat{R}(u_i), \tilde{\mathcal{T}}^N)| \le 2\eta_{\max}^\Delta .$$

With the assumption that the keys $\{u_i\}_{i \in [N]}$ are pairwise distinct, the ranks $(R(u_i))_{i \in [N]}$ form a permutation of $[N]$.

Given that $|R(u_k) - \widehat{R}(u_k)| \le \eta_{\max}^\Delta$ for all $k \in [N]$, it holds for any $i, j \in [N]$ that

$$\begin{aligned}
\widehat{R}(u_j) \le \widehat{R}(u_i) &\implies R(u_j) - \eta_{\max}^\Delta \le R(u_i) + \eta_{\max}^\Delta \\
&\implies R(u_j) \le R(u_i) + 2\eta_{\max}^\Delta ,
\end{aligned}$$

hence, given that $\mathcal{T}^N = \{\widehat{R}(u_j)\}_{j \in [N]}$ and by definition (1) of the rank $r(\widehat{R}(u_i), \mathcal{T}^N)$, we have

$$\begin{aligned}
r(\widehat{R}(u_i), \mathcal{T}^N) &= \#\{j \in [N] : \widehat{R}(u_j) \le \widehat{R}(u_i)\} \\
&\le \#\{j \in [N] : R(u_j) \le R(u_i) + 2\eta_{\max}^\Delta\} \\
&= \#\{k \in [N] : k \le R(u_i) + 2\eta_{\max}^\Delta\} \tag{9} \\
&= \min(N, \, R(u_i) + 2\eta_{\max}^\Delta) \\
&\le R(u_i) + 2\eta_{\max}^\Delta , \tag{10}
\end{aligned}$$

where Equation 9 holds because $(R(u_i))_{i \in [N]}$ is a permutation of $[N]$. Similarly, we have for all $i, j \in [N]$ that

$$\begin{aligned}
R(u_j) \le R(u_i) - 2\eta_{\max}^\Delta &\implies R(u_j) + \eta_{\max}^\Delta \le R(u_i) - \eta_{\max}^\Delta \\
&\implies \widehat{R}(u_j) \le \widehat{R}(u_i) ,
\end{aligned}$$

and it follows for all $i \in [N]$ that

$$\begin{aligned}
r(\widehat{R}(u_i), \mathcal{T}^N) &= \#\{j \in [N] : \widehat{R}(u_j) \le \widehat{R}(u_i)\} \\
&\ge \#\{j \in [N] : R(u_j) \le R(u_i) - 2\eta_{\max}^\Delta\} \\
&= \#\{k \in [N] : k \le R(u_i) - 2\eta_{\max}^\Delta\} \\
&= \max(0, \, R(u_i) - 2\eta_{\max}^\Delta) \\
&= R(u_i) - 2\eta_{\max}^\Delta . \tag{11}
\end{aligned}$$

From (10) and (11), we deduce that

$$\forall i \in [N]: \quad |R(u_i) - r(\widehat{R}(u_i), \mathcal{T}^N)| \le 2\eta_{\max}^\Delta ,$$

Combining this with (8) and (7) yields the wanted result. $\qquad\square$

**Proof of Theorem 3.3**

*Proof.* When a new key $u_i$ is to be inserted, it is first inserted in the dynamic vEB tree $\mathcal{T}^i$ with integer key $\widehat{R}(u_i)$, and its predecessor $\hat{w}$ in $\mathcal{T}^i$ is retrieved. These first operations require $O(\log \log N)$ time. The position of $u_i$ in $\mathcal{Q}^i$ is then obtained via an exponential search starting from $\hat{w}$, which requires $O(\log |r(u_i, \mathcal{Q}^i) - r(\hat{w}, \mathcal{Q}^i)|)$ expected time by Theorem 3.1. Finally, inserting $u$ in the found position takes expected $O(1)$ time.

For any newly inserted element, by accounting for the potential future deletion time via ExtractMin at the moment of insertion, we can ensure that all ExtractMin operations require constant amortized time. This approach results in only an additional $O(\log \log N)$ time for insertions.

Therefore, to prove Theorem 3.3, we only need to show that $\log |r(u_i, \mathcal{Q}^i) - r(\hat{w}, \mathcal{Q}^i)| = O(\log \max_{j \in [N]} \eta^\Delta(u_i))$. Since $\hat{w}$ is the predecessor of $u_i$ in $\mathcal{T}^i$, it holds that $r(\widehat{R}(u_i), \mathcal{T}^i) \in \{r(\widehat{R}(\hat{w}), \mathcal{T}^i), r(\widehat{R}(\hat{w}, \mathcal{T}^i) + 1\}$, the first case occurs if $\widehat{R}(u_i) = \widehat{R}(\hat{w})$, and the second if $\widehat{R}(u_i) > \widehat{R}(\hat{w})$. Using this observation and Lemma C.2, it follows that

$$
\begin{aligned}
&|r(u_i, \mathcal{Q}^i) - r(\hat{w}, \mathcal{Q}^i)| \\
&\leq |r(u_i, \mathcal{Q}^i) - r(\widehat{R}(u_i), \mathcal{T}^i)| + |r(\widehat{R}(u_i), \mathcal{T}^i) - r(\widehat{R}(\hat{w}), \mathcal{T}^i)| + |r(\hat{w}, \mathcal{Q}^i) - r(\widehat{R}(\hat{w}), \mathcal{T}^i)| \\
&\leq 4 \max_{j \in [N]} \eta^\Delta(u_j) + 1 \,,
\end{aligned}
$$

and it follows that $\log |r(u_i, \mathcal{Q}^i) - r(\hat{w}, \mathcal{Q}^i)| = O(\log \max_{j \in [N]} \eta^\Delta(u_i))$, which concludes the proof. $\square$

# D   Lower bounds

A priority queue can be used for sorting a sequence $A = (a_i)_{i \in [n]} \in \mathcal{U}^n$, by first inserting all the elements in the priority queue, then repeatedly extracting the minimum until the priority queue is empty. In settings with dirty comparisons or *positional* predictions, the number of comparisons required by this sorting algorithm is constrained by the impossibility result demonstrated in Theorem 1.5 of Bai and Coester [2023]. We use this impossibility result to prove the lower bounds stated in Theorem 3.4.

## D.1   Impossibility result for sorting with predictions

We begin in this section by summarizing the setting and the impossibility result demonstrated in Theorem 1.5 of Bai and Coester [2023] for sorting with predictions.

**Positional predictions**   In the *positional prediction* model, the objective is to sort a sequence $A = (a_i)_{i \in [n]}$, given a prediction $\widehat{R}_i \in [n]$ of $R_i = r(a_i, A)$ for all $i \in [n]$. This model differs from our rank prediction model in that the sequence $A$ and the predictions $\widehat{R} = (\widehat{R}_i)_{i \in [n]}$ are given offline to the algorithm. Two different error measures are considered in Bai and Coester [2023], but we restrict ourselves to the *displacement error* $\eta_i^\Delta = |r(a_i, A) - \widehat{R}_i|$, which is the same as our rank prediction error 4. In all the following, consider that $\widehat{R}$ is a fixed permutation of $[n]$, i.e. the predicted ranks are pairwise distinct. For all $\xi \geq 1$, consider the following set of instances

$$
\text{cand}(\hat{R}, n\xi) = \{A \in \mathcal{U}^n : \sum_{i=1}^n \log(\eta_i^\Delta + 2) \leq n\xi\} \,.
$$

The authors of Bai and Coester [2023] prove that, if $1 \leq \xi \leq O(\log n)$, then no algorithm can sort every instance from $\text{cand}(\hat{R}, n\xi)$ with $o(n\xi)$ comparisons. However, in their proof, they demonstrate a stronger result: no algorithm can sort every instance from $\mathcal{I}_n(\widehat{R}, \xi)$ with $o(n\xi)$ comparisons, where $\mathcal{I}_n(\widehat{R}, \xi)$ is the subset of $\text{cand}(\hat{R}, n\xi)$ defined by

$$
\mathcal{I}_n(\widehat{R}, \xi) = \{A \in \mathcal{U}^n : \max_{i \in [n]} \log(\eta_i^\Delta + 2) \leq \xi\} \,. \tag{12}
$$

**Dirty comparisons**   in the dirty comparison model, the authors prove an analogous result by a reduction to the positional prediction model. More precisely, any permutation $\widehat{R}$ on $[n]$ defines a unique dirty order $\widehat{<}$ on $A$ given by $a_i \mathbin{\widehat{<}} a_j \iff \widehat{R}_i < \widehat{R}_i$, and $\max_{i \in [n]} \eta_i \leq \max_{i \in [n]} \eta_i^\Delta$, where $\eta_i = \eta(a_i, A) = \#\{j \in [n] : (a_i \mathbin{\widehat{<}} a_j) \neq (a_i < a_j)\}$. We deduce that

$$\mathcal{I}_n(\widehat{R}, \xi) \subset \{A \in \mathcal{U}^n : \max_{i \in [n]} \log(\eta_i + 2) \leq \xi\} . \tag{13}$$

Hence, given the dirty order $\widehat{<}$, there is no algorithm that can sort every instance with $\max_{i \in [n]} \log(\eta_i + 2) \leq \xi$ in $o(n\xi)$ time.

In the following, we use these lower bounds on sorting to prove our Theorem 3.4.

### D.2   Pointer prediction model

For any permutation $\pi$ of $[n]$ and $i \in [n]$, we denote by $A_i^\pi = \{a_{\pi(j)}\}_{j \in [i]}$. This first elementary Lemma uses ideas from the proof of Theorem 1.3 in Bai and Coester [2023].

**Lemma D.1.** *A permutation $\pi$ of $[n]$ satisfying $\widehat{R}_{\pi(1)} \leq \ldots \leq \widehat{R}_{\pi(n)}$ can be constructed in $O(n)$ time, and it holds for all $i \in \{2, \ldots, n\}$ that*

$$|r(a_{\pi(i)}, A_{i-1}^\pi) - r(a_{\pi(i-1)}, A_{i-1}^\pi)| \leq \eta_{\pi(i-1)}^\Delta + \eta_{\pi(i)}^\Delta + \widehat{R}_{\pi(i)} - \widehat{R}_{\pi(i-1)} .$$

*Proof.* The elements of $A$ can be sorted in non-decreasing order of their predicted ranks $\widehat{R}_i$ within $O(n)$ time using a bucket sort. Hence, we obtain the permutation $\pi$. For all $i \in \{2, \ldots, n\}$, $|r(a_{\pi(i)}, A_{i-1}^\pi) - r(a_{\pi(i-1)}, A_{i-1}^\pi)|$ is the number of elements in $A_{i-1}^\pi$ whose ranks are between those of $a_{\pi(i)}$ and $a_{\pi(i-1)}$. This is at most the number of elements in $A$ whose ranks are between those of $a_{\pi(i)}$ and $a_{\pi(i-1)}$, which is $|R_{\pi(i)} - R_{\pi(i-1)}|$. We deduce that

$$\begin{aligned}
|r(a_{\pi(i)}, A_i^\pi) - r(a_{\pi(i-1)}, A_i^\pi)| &\leq |R_{\pi(i)} - R_{\pi(i-1)}| \\
&\leq |R_{\pi(i)} - \widehat{R}_{\pi(i)}| + |R_{\pi(i-1)} - \widehat{R}_{\pi(i-1)}| + |\widehat{R}_{\pi(i)} - \widehat{R}_{\pi(i-1)}| \\
&= \eta_{\pi(i-1)}^\Delta + \eta_{\pi(i)}^\Delta + \widehat{R}_{\pi(i)} - \widehat{R}_{\pi(i-1)} ,
\end{aligned}$$

where we used the triangle inequality and that $\widehat{R}_{\pi(i)} \geq \widehat{R}_{\pi(i-1)}$.

$\square$

We move now to the proof of our lower bound in the pointer prediction model, by reducing the problem of sorting with positional predictions to the design of a learning-augmented priority queue with pointer predictions.

### D.2.1   Proof of Theorem 3.4

*Proof.* Assume that there exists an implementation of a priority queue $\mathcal{Q}$ augmented with pointer predictions, supporting ExtractMin with $O(1)$ comparisons and the insertion of any new key $u$ with $o(\log \vec{\eta}(u, \mathcal{Q}))$ comparisons. This means that, regardless of the history of operations made on $\mathcal{Q}$, the number of comparisons used by ExtractMin is at most a constant $C$, and for any $\xi \geq 1$, inserting a key $u$ such that $\log(\vec{\eta}(u, \mathcal{Q}) + 2) \leq \xi$ requires at most $\varepsilon(\xi)\xi$ comparisons, where $\varepsilon(\cdot)$ is a positive function satisfying $\lim_{\xi \to \infty} \varepsilon(\xi) = 0$. We will show that the existence of such a priority queue contradicts the impossibility result of sorting with positional predictions.

Let $1 \leq \xi \leq O(\log n)$, $\widehat{R}$ a fixed permutation of $[n]$, and $A$ an arbitrary instance from the set $\mathcal{I}_n(\widehat{R}, \xi)$ defined in (12). Let $\pi$ be the permutation satisfying the property of Lemma D.1, and consider the sorting algorithm which inserts the elements of $A$ in $\mathcal{Q}$ in the order given by $\pi$, then extracts the minimum repeatedly until $\mathcal{Q}$ is emptied. Let us denote by $\mathcal{Q}^i$ the state of the $\mathcal{Q}$ after $i$ insertions. Upon the insertion of $a_{\pi(i)}$, the algorithm uses $\widehat{\text{Pred}}(a_{\pi(i)}, \mathcal{Q}^{i-1}) = a_{\pi(i-1)}$ as a pointer prediction. By (3), the error of this prediction is

$$\vec{\eta}(a_{\pi(i)}, \mathcal{Q}^{i-1}) = |r(a_{\pi(i)}, A_{i-1}^\pi) - r(a_{\pi(i-1)}, A_{i-1}^\pi)| ,$$

where $A_{i-1}^{\pi} = \{a_{\pi(j)}\}_{j<i}$, and by Lemma D.1, we have that

$$\vec{\eta}(a_{\pi(i)}, \mathcal{Q}^{i-1}) \leq \eta_{\pi(i-1)}^{\Delta} + \eta_{\pi(i)}^{\Delta} + \widehat{R}_{\pi(i)} - \widehat{R}_{\pi(i-1)} . \tag{14}$$

$\widehat{R}$ is a permutation of $[n]$, hence $\widehat{R}_{\pi(i)} = \widehat{R}_{\pi(i-1)} + 1$ by definition of $\pi$. Moreover, $A \in \mathcal{I}_n(\xi)$, thus $\max(\log(\eta_{\pi(i-1)}^{\Delta} + 2), \log(\eta_{\pi(i)}^{\Delta} + 2)) \leq \xi$, and it follows that

$$\begin{aligned}
\log(\vec{\eta}(a_{\pi(i)}, \mathcal{Q}^{i-1}) + 2) &\leq \log(\eta_{\pi(i-1)}^{\Delta} + \eta_{\pi(i)}^{\Delta} + 3) \\
&\leq \log(\eta_{\pi(i-1)}^{\Delta} + 2) + \log(\eta_{\pi(i)}^{\Delta} + 2) \\
&\leq 2\xi ,
\end{aligned}$$

where the second inequality is a consequence of $\log(\alpha + \beta) \leq \log(\alpha) + \log(\beta)$ for all $\alpha, \beta \geq 2$.

Therefore, all the insertions into $\mathcal{Q}$ require at most $(2\xi \cdot \varepsilon(2\xi))$ comparisons, and the total number of comparisons $T$ used to sort $A$ is at most

$$\begin{aligned}
T &\leq n \left(2\xi \cdot \varepsilon(2\xi) + C\right) \\
&= n\xi \left(2 \cdot \varepsilon(2\xi) + \frac{C}{\xi}\right) \\
&= o(n\xi) .
\end{aligned}$$

This means that any instance in $\mathcal{I}_n(\xi)$ can be sorted using $o(n\xi)$ comparisons, which contradicts the lower bound on sorting algorithms augmented with positional predictions. $\qquad\square$

### D.2.2 Rank prediction and dirty comparisons model

In the rank prediction model, the total number of inserted keys is $N = n$. Denote by $\eta_i = \eta(a_i, A)$ for all $i \in [n]$. If there is a data structure $\mathcal{Q}$ not satisfying the lower bound of Theorem 3.4 for positional predictions, then we can use it for sorting $A$. Let $1 \leq \xi \leq O(\log n)$. Similarly to the proof for pointer predictions, if $\widehat{R}$ is a permutation of $[n]$, using $\mathcal{Q}$ for sorting any instance $A \in \mathcal{I}_n(\xi)$ requires at most $T = o(n\xi)$ comparisons, which contradicts the lower bound on learning-augmented sorting algorithms.

The same arguments, combined with (13), give the result also for the dirty comparison model.

## E  Applications

### E.1  Sorting

In the case where the algorithm is given offline access to the sequence $A = (a_1, \ldots, a_n)$ to sort and to the rank predictions $(\widehat{R}(a_i))_{i \in [n]}$, we argue that pointer predictions can be used to sort the sequence within a $O(\sum_{i \in [n]} \log(\eta_i^{\Delta} + 2))$.

With offline rank predictions, by Lemma D.1, the algorithm can construct a permutation $\pi$ of $[n]$ satisfying $\widehat{R}(a_{\pi(1)}) \leq \ldots \leq \widehat{R}(a_{\pi(n)})$, and we showed in the proof of Theorem 3.4, in (14), that inserting the elements of $A$ into the priority queue in the order given by $\pi$, then taking each inserted element $a_{\pi(i)}$ as pointer prediction for the following one $a_{\pi(i+1)}$, yields a pointer prediction error of

$$\vec{\eta}(a_{\pi(i)}, \mathcal{Q}^{i-1}) \leq \eta_{\pi(i-1)}^{\Delta} + \eta_{\pi(i)}^{\Delta} + \widehat{R}_{\pi(i)} - \widehat{R}_{\pi(i-1)}$$

for all $i \in \{2, \ldots, n\}$. By Theorem 3.1, the runtime for inserting $a_{\pi(i)}$ using this pointer prediction is $O(\log \vec{\eta}(a_{\pi(i)}, \mathcal{Q}^{i-1}))$ . The total time for inserting all the elements into the priority queue is

therefore at most proportional to

$$\sum_{i=2}^{n} \log(\vec{\eta}(a_{\pi(i)}, \mathcal{Q}^{i-1}) + 2) \leq \sum_{i=2}^{n} \log(\eta_{\pi(i-1)}^{\Delta} + \eta_{\pi(i)}^{\Delta} + \widehat{R}_{\pi(i)} - \widehat{R}_{\pi(i-1)} + 2)$$

$$\leq \sum_{i=2}^{n} \log((\eta_{\pi(i-1)}^{\Delta} + 2) + (\eta_{\pi(i)}^{\Delta} + 2) + (\widehat{R}_{\pi(i)} - \widehat{R}_{\pi(i-1)} + 2))$$

$$\leq \sum_{i=2}^{n} \left( \log(\eta_{\pi(i-1)}^{\Delta} + 2) + \log(\eta_{\pi(i)}^{\Delta} + 2) + \log(\widehat{R}_{\pi(i)} - \widehat{R}_{\pi(i-1)} + 2)) \right)$$

$$\leq 2\sum_{i=1}^{n} \log(\eta_{\pi(i)}^{\Delta} + 2) + \sum_{i=2}^{n} (\widehat{R}_{\pi(i)} - \widehat{R}_{\pi(i-1)}) + n$$

$$\leq 2\sum_{i=1}^{n} \log(\eta_{\pi(i)}^{\Delta} + 2) + \widehat{R}_{\pi(2)} + n$$

$$= O\left( \sum_{i \in [n]} \log(\eta_i^{\Delta} + 2) \right).$$

The third inequality results from inequation $\log(\alpha + \beta + \gamma) \leq \log \alpha + \log \beta + \log \gamma$ for all $\alpha, \beta, \gamma \geq 2$, and the subsequent inequality follows from $\log(\alpha + 1) \leq \alpha$ for all $\alpha \geq 0$. Finally, to complete the sorting algorithm, the minimum is repeatedly extracted from the priority queue until is it empty, and each ExtractMin only takes $O(1)$ time in the skip list.

### E.2   Dijkstra's algorithms

We give in this section more details on the complexity of Dijkstra's algorithm using our priority queue augmented with rank predictions.

Consider any priority queue implementation, having a time complexity $T_{\text{Insert}}$ for insertion, $T_{\text{ExtractMin}}$ for extracting the minimum, and $T_{\text{DecreaseKey}}$ for decreasing the key of an element. There are two possible implementations of Dijkstra's algorithm with priority queues. The first one only uses the operations Insert and ExtractMin, and the maximum number of inserted elements is $m + 1$, which yields a complexity of $O(mT_{\text{Insert}} + mT_{\text{ExtractMin}})$. The second implementation, using also the DecreaseKey operation, yields a complexity of $O(n(T_{\text{Insert}} + T_{\text{ExtractMin}}) + mT_{\text{DecreaseKey}})$.

Recalling that the number of edges is at most $n^2$, using a binary heap in any of both implementations gives a total runtime of $O(m \log n)$. On the other hand, using a Fibonacci heap in the second implementation yields a runtime of $O(n \log n + m)$.

In the rank predictions model, we consider that the priority queue only uses Insert and ExtractMin operations, hence we use the first implementation. Denoting by $\{d_i\}_{i \in [m]}$ the $m$ distinct keys that are inserted, these keys are only revealed online to the priority queue. If they are accompanied by predictions $(\widehat{R}(d_i))_i$ of their ranks, then each insertion requires $O(\log \log m + \log(\max_{i \in [m+1]} |R(d_i) - \widehat{R}(d_i)| + 2))$ amortized time and $O(\log \max_{i \in [m+1]} |R(d_i) - \widehat{R}(d_i)|)$ comparisons, while extractions only require a constant amortized time each. The total runtime is therefore

$$O\big(m \log \log n + m \log \max_{i \in [m+1]} |R(d_i) - \widehat{R}(d_i)|\big),$$

and the total number of comparisons is $O(m \log(\max_{i \in [m+1]} |R(d_i) - \widehat{R}(d_i)|))$.

## F   Additional experiments

### F.1   Sorting

The problem of sorting with similar prediction models have been studied in Bai and Coester [2023], hence we numerically compare sorting using our learning-augmented priority queue (LAPQ) with their sorting algorithms. To run their algorithms, we used the code provided by Bai and Coester

[2023] in their paper. We give here additional experimental results for the *class* and the *decay* setting, for smaller values of $n$.

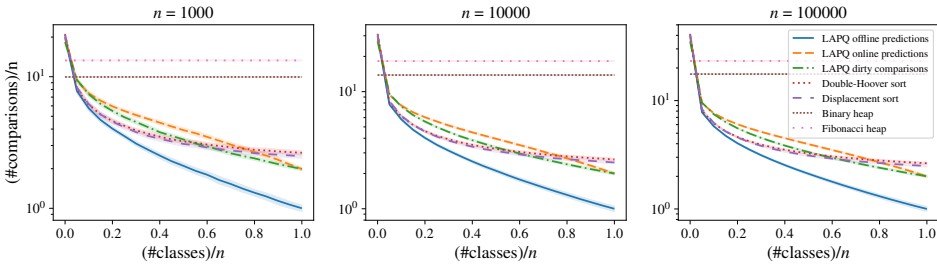

Figure 6: Sorting with rank predictions in the *class* setting, for $n \in \{1000, 10000, 100000\}$.

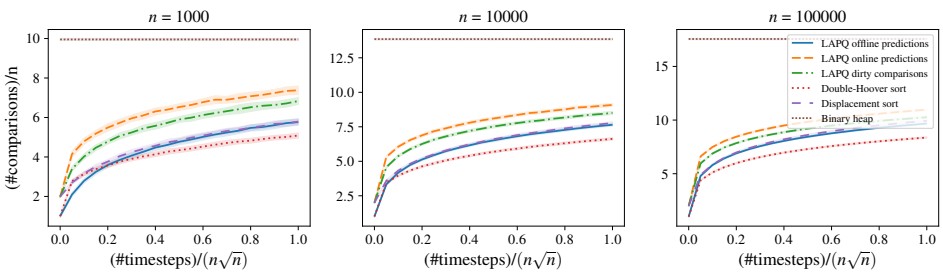

Figure 7: Sorting with rank predictions in the *decay* setting, for $n \in \{1000, 10000, 100000\}$.

## F.2  Dijkstra's algorithm

Considering the same experimental setting presented in Section 5, Figures 8 and 9 show the obtained results for the cities of Brussels, Paris, New York, and London, which have. The numbers of nodes $n$ and edges $m$ in each city graph are indicated in the figures.

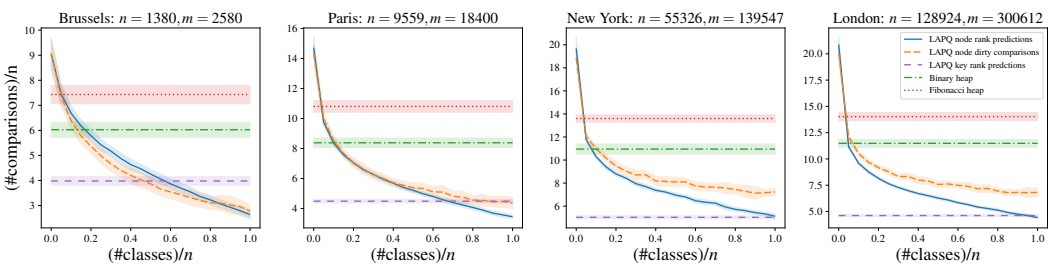

Figure 8: Dijkstra's algorithm on city maps with class predictions

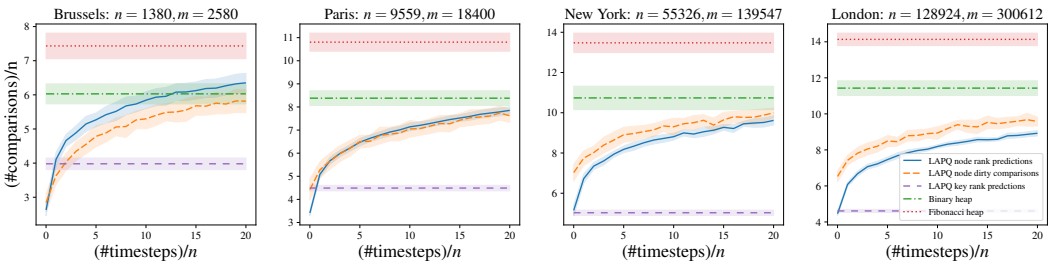

Figure 9: Dijkstra's algorithm on city maps with decay predictions

**Simulations on Poisson Voronoi Tesselations**  We also evaluate the performance of Dijkstra's algorithm with our LAPQ on synthetic random graphs. For this, we use Poisson Voronoi Tessellations (PVT), which are a commonly used random graph model for street systems Gloaguen et al. [2006], Gloaguen and Cali [2018], Benomar et al. [2022a,b].

PVTs are random graphs created by sampling an integer $N$ from a Poisson distribution with parameter $n \geq 1$. Subsequently, $N$ points, termed "seeds," are uniformly chosen at random within a two-dimensional region $I$, typically $[0, 1]^2$. A Voronoi diagram is then generated based on these seeds.

This process results in a planar graph where edges represent the boundaries between the cells of the Voronoi diagram, and the nodes are their intersections. For $I = [0, 1]^2$, the expected number of nodes in this construction is $n$.

Figure 10 provides a visualization of a PVT with $n = 100$.

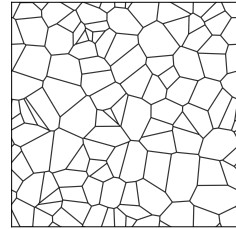

Figure 10: PVT with $n = 100$

We present the results in the class and the decay settings respectively in Figures 11 and 12. Similar to previous experiments with Dijkstra on city maps, these figures illustrate how the number of comparisons decreases when the LAPQ is augmented with *node rank* predictions or with the corresponding dirty comparator. We compare them with the number of comparisons induced by using a binary or Fibonacci heap, as well as with the number of comparisons of the LAPQ augmented with *key rank predictions*.

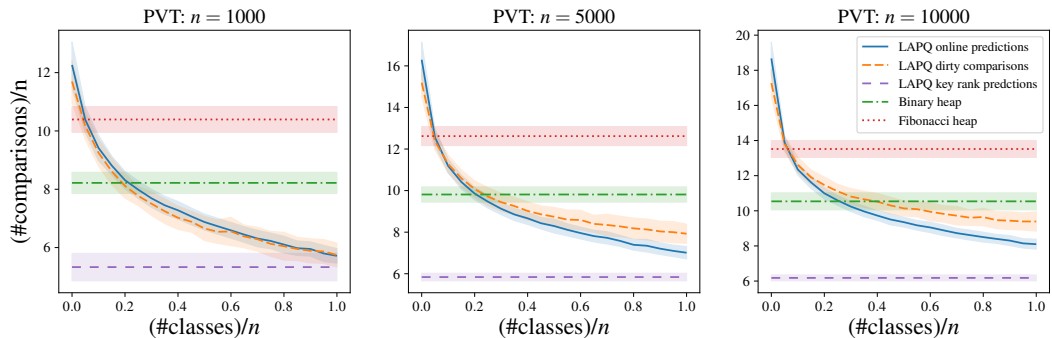

Figure 11: Dijkstra's algorithm on Poisson Voronoi Tesselation with class predictions

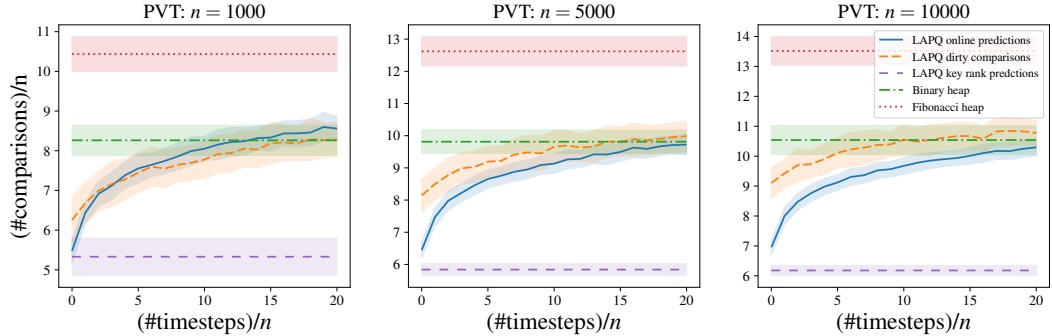

Figure 12: Dijkstra's algorithm on Poisson Voronoi Tesselation with decay predictions

The same observations regarding the performance improvement can be made, as in the previous experiments with city maps. However, in PVT tessellations, the performance of the LAPQ with key rank predictions surpasses even that of perfect node rank predictions. This is due to the PVTs having a more uniform structure across space.

