# OpenReview forum: "Learning-Augmented Priority Queues"
_NeurIPS.cc/2024/Conference — NeurIPS 2024 poster_

### Official Review · Reviewer_KF5a · 2024-07-12

**Soundness:** 2
**Presentation:** 2
**Contribution:** 2
**Rating:** 4
**Confidence:** 4

**Summary:**

The study investigates the integration of learning-augmented frameworks into the design of priority queues, focusing on enhancing worst-case performance using potentially inaccurate predictions. It examines three specific prediction models—dirty comparisons, pointer prediction, and rank prediction—applied within priority queues based on skip lists for sorting and Dijkstra algorithms. The research demonstrates how these predictive models can effectively optimize priority queue operations. Furthermore, it establishes the optimality of
the proposed solution and explores potential real-world applications of the findings.

**Strengths:**

S1: The mathematical proof process of the paper is highly rigorous, demonstrating strong
credibility.
S2: The methods proposed in the paper significantly reduce the number of data item
comparisons during the internal execution of basic operations in priority queues, thus
possessing the potential to pique the interest of researchers in related fields.

**Weaknesses:**

O1: Writing Quality and Readability Issues
O1.1 Poor logical flow and transitions: Before discussing the learning-enhanced framework, the limitations and necessity of traditional priority queue methods are not sufficiently explained, leading to unnatural logical transitions. For instance, after stating in line 21, "However, it is established that a priority queue with n elements cannot guarantee o(log n) time for all the required operations," there should be a more detailed explanation of why this is a significant limitation. Similarly, after line 158's statement, "To improve the
insertion complexity with dirty comparisons, we first tackle the search problem in this setting," the paper jumps directly into technical implementation without a clear bridging sentence explaining why this approach can improve complexity.

O1.2 Lengthy sentences unsuitable for academic writing: Some sentences are overly lengthy, which does not meet the standards of academic writing. For example, in line 163, "Indeed,...."

O1.3 Dense concepts with insufficient explanation: The paper introduces many concepts but fails to adequately explain some important ones. Although many specific examples of priority queue applications are provided, these details could be more credibly summarized and
simplified to make the paper more concise. Detailed explanations of application cases might belong in the "Background" section rather than the "Introduction."

O1.4 Figures not referenced or adequately explained in the text: The paper contains only three figures, but they are not referenced in the text. For instance, Figure 1 on skip lists is neither clearly explained in the text nor supplemented with detailed captions. Additionally,
the figure does not illustrate the dynamic process of finding and deleting the minimum value in a priority queue using a skip list.

O1.5 Lack of summaries and overviews: There is no summary of the innovations, contributions, or experimental results in the Introduction section.

O2: Lack of detailed motivation and innovation overview. The paper does not provide a detailed introduction to the motivation behind the work or summarize its innovative aspects. Although the authors point out the limitation of priority queues in guaranteeing O(logn)
operations in the worst case on line 21, they do not explain in which application scenarios this limitation occurs. Appendix A also fails to discuss whether related works have addressed this issue and what shortcomings remain.

O3: Dependence on previous work with insufficient innovation. This paper builds on a previous work "Xingjian Bai and Christian Coester. Sorting with predictions. Advances in Neural Information Processing Systems, 36, 2023.", requiring readers to have prior knowledge of the previous paper to understand much of its content. However, this paper lacks innovative aspects compared to the previous one, merely combining several prediction algorithms (Dirty Comparison, rank prediction, pointer prediction) with priority queues.

O4: Insufficient comparative analysis of algorithm bounds. While the authors provide numerous bounds for their algorithms in the main text and appendices, these proofs are convincing but lack comparison with related works and original priority queues regarding
these bounds. The intrinsic relationship between the designed algorithms and these bounds is not clearly explained. I suggest that the authors provide a comparative table to illustrate the advantages of various bounds more clearly and detail how their algorithms ensure these bounds during the algorithm introduction.

O5: Issues in the experimental section. The experimental section introduces important settings, such as class/decay setting, without prior explanation. These settings are suddenly presented in the experimental part with vague explanations, requiring readers to refer to
cited articles mentioned in O3 for understanding. As key experimental variables, these settings should be formally defined and described. The authors also fail to explain the specific application scenarios corresponding to these settings and their differences.

O6: Experimental setup issues. The experimental setup has significant issues. Merely using the number of comparisons as a metric is inadequate. The authors should also record the error metrics for the three algorithms mentioned in Section 1.1 in their experiments,
including the number of dirty comparisons, pointer prediction errors, and rank prediction errors. Furthermore, the experiments should compare various methods regarding actual execution time and memory usage in sorting and Dijkstra algorithms. I recommend that the
authors refer to relevant experimental metrics used in research papers from the learned index field.

O7: Unsubstantiated claims about adversarial insertion order. In line 110, the authors mention that their proposed method has advantages under adversarial insertion orders, but they neither explain which application scenarios would encounter this situation nor provide
experimental results to validate this claim.

**Questions:**

Q1: Could you provide more specific algorithms that utilize priority queues and indicate under what conditions the original design might experience degradation during basic operation execution? This question pertains to the motivation behind this study.

Q2: Could you provide a more detailed explanation of the class / decay setting (O5)? Particularly, regarding the class setting, as (#classes)/n increases, the number of partitions increases, leading to fewer comparisons. What is the basis for such experimental settings?

Q3: Where does the trade-off lie in your algorithm? Is there a need to allocate additional space for the prediction models? It seems you have omitted necessary discussions on space complexity and have not observed memory usage in your experiments.

---

> ### Author Rebuttal · Authors · 2024-08-05
>
> We thank the reviewer for their feedback and suggestions. We address below their concerns.
> ### Weaknesses
> * **O1.1. (Limitations of standard priority queues, line 21)** The lack of $o(\log n)$ time priority queues is an impossibility result, hence a limitation. We are not quite sure what type of detailed explanation the reviewer is requesting.
> * **O1.1. (Sentence in line 158)** Immediately before that sentence, in the same paragraph, we explain that a heap can use a binary search algorithm for insertion. Having a better search algorithm, using predictions, would therefore immediately improve the insertion complexity.
> * **O1.2.** We kindly disagree regarding line 163. That sentence is not overly complex, and we do not think this constitutes a weakness in the paper.
> * **O1.3.** Could the reviewer specify which concepts they would like us to explain more? The list of applications in the first paragraph illustrates their ubiquity, but the details are not important.
> * **O1.4.** We will add explicit references to Figures 1 and 2 in the text. Figure 1 is merely an illustration of a skip list and corresponds to the description above it. Figure 2 includes a clear caption and corresponds to the discussion to its left. As noted in line 357, a detailed discussion of Figures 3 and 4 is included in Appendix G, which we could not include in the main body of the paper due to space limitations.
> * **O1.5.** The section "our results" summarizes our main contributions and experimental results.
> * **O2.** We address all the points raised here in O1.5, O1.1 and Q1.
> * **O3.** The paper does not require any prior knowledge of the previous work by Bai and Coester for comprehension. We study, among others, the dirty comparison model they have introduced. However, we formally define the model and explain its relevance for priority queues. In Section 4, we demonstrate how their results can be derived as corollaries of ours. For sorting, we also employ their experimental setup to compare with their sorting algorithms, which only proves the efficiency of our priority queue. We use their results only to prove the lower bound in the pointer prediction model.
> * **O4.** The complexities of priority queues are detailed in the related work, and are recalled multiple times throughout the paper (lines 85, 149, 156, 198, ...). However, as suggested by the reviewer, we will add a table in the section "our results" to summarize the different bounds.
> * **O5.** The motivation behind these settings is not discussed in sufficient detail due to space limitations. Their relevance to Dijkstra's algorithm is addressed in Appendix G. If the paper is accepted, we will use the additional page to provide more detailed information on the motivation (see also our reponse to Q2 below).
> * **O6.** The prior work of Bai and Coester, which we can compare to, used number of comparisons as performance metric. This has the advantage that results are replicable independent of hardware and implementation details. Therefore, we opted to use the number of comparisons as performance metric as well. In particular, this allowed us to use a simple Python implementation for our algorithms, whereas for the algorithms by Bai and Coester we used their existing C++ implementation. The prediction error in the three models is correlated to the number of classes in the class setting, and to the number of timesteps in the decay setting. Thus, the x-axis in the figures represents the amount of perturbation in the predictions. The works in the learned index field that we are aware of have a much stronger experimental focus, whereas our paper's primary focus is theoretical. Many other experiments are possible, which could be part of a separate study.
> * **O7.** The bounds we establish for sorting hold for any insertion order, even if chosen by an adversary. This contrasts with the algorithm of Bai and Coester, which requires a random insertion order. So our algorithms are defined in a strictly broader setting. Application scenarios of the broader setting we capture are any situations where a sorted order must be maintained while items are added (and possibly deleted) over time, rather than being known upfront. Validating this claim experimentally would not make sense, as the algorithms of Bai and Coester cannot process such inputs.
>
> ### Questions
> * **Q1.** Regardless of the use case, the priority queue operations with a binary heap always require $\Theta(\log n)$ comparisons. In a skip-list, insertion always requires expected $\Theta(\log n)$ comparisons. The motivation of the paper is to use predictions to reduce these complexities.
>  * **Q2.** The formal description of both settings is given in the experiments section (line 337). In the context of sorting, items often have grades that provide a partial ordering. For instance, students might have GPA grades A,B,C,..., and our goal is to determine their precise ranking. The decay setting models situations where an initial ordering of items may have evolved over time. The perturbation of the ordering depends on the time elapsed between the initial ranking and the current time step. The relevance of both settings for Dijkstra's algorithm is given in Appendix G (line 986). If the paper is accepted, we will use the extra page to expand this discussion in the experiments section.
> * **Q3.** As is common in the literature on algorithms with predictions, we treat the ML models delivering the predictions as a black box. Their space consumption depends on the implementation of the model. The expected space occupied by a skip list is $O(n)$, and in the case of rank predictions, an additional $O(N)$ space is needed to store the vEB tree.

---

> > ### Comment · Reviewer_KF5a · 2024-08-11
> > **Thank you for your clarification**
> >
> > Thank you for your detailed response. However, I feel my concerns regarding the experiments remain inadequately addressed, which is a significant issue for me in this paper. I will maintain my recommendation.

---

> > > ### Author Response · Authors · 2024-08-12
> > >
> > > Thanks for your response. From the reply, we gather that it seems to be primarily O6 and possibly O5 that the reviewer remains concerned about.
> > >
> > > We appreciate that the reviewer’s preference may be for papers with detailed experimental analyses, as is common in many fields of research. For the field of algorithms-with-predictions (and theoretical works more generally), the type of experiments that the reviewer is suggesting — reporting on memory usage etc — would be extremely unusual though (unless that is the quantity being optimized). In fact, many similarly flavored algorithms-with-predictions papers don’t include any experiments at all, with some examples from last year’s NeurIPS and ICML being:
> > >
> > > [1] NeurIPS’23: Li et al. “Beyond Black-Box Advice: Learning-Augmented Algorithms for MDPs with Q-Value Predictions”
> > > [2] NeurIPS’23: Balcan et al. “Bicriteria Multidimensional Mechanism Design with Side Information”
> > > [3] ICML’23: Antoniadis et al. “Mixing Predictions for Online Metric Algorithms”
> > > [4] ICML’23: Lassota et al. “Minimalistic Predictions to Schedule Jobs with Online Precedence Constraints”
> > > [5] ICML’23: Antoniadis et al. “Paging with Succinct Predictions”
> > >
> > > Those that do include experiments typically restrict them to a brief section at the end, with similar setup choices to us, that merely serve as additional support to the prior main results, rather than being main results themselves.
> > >
> > > But there’s another, more important reason why using running times as a performance measure would actually be misleading in our case: The motivation for the dirty comparison setting is to have two types of comparisons: slow exact comparisons and fast inexact comparisons. An example could be comparing molecule structures for a potential vaccine, where slow comparisons would involve lengthy clinical trials, while fast comparisons return imprecise results very cheaply. In our experiments, however, the simulation of either type of comparison is equally very fast, so running time wouldn’t capture the true performance at all. One could address this by having the simulation artificially wait for a long time whenever it performs an exact comparison. But this is essentially exactly what we achieve by using the number of clean comparisons as a performance measure. Similarly for the other prediction types, in reality there’s a flexible overhead for producing predictions that depends on the prediction model and which the simulations wouldn’t capture. The point of algorithms-with-predictions is to study the effective usage of predictions separately from their generation. For these reasons, and the ones mentioned earlier, comparison complexity is the better measure of performance for the problems we consider, while the running time of simulations would be rather meaningless.

---

### Official Review · Reviewer_unQV · 2024-07-12

**Soundness:** 4
**Presentation:** 3
**Contribution:** 3
**Rating:** 7
**Confidence:** 4

**Summary:**

The paper studies various beyond worst-case models for priority queues, a fundamental data structure. A learning-augmented viewpoint is taken and the authors comprehensively study three different natural prediction models: dirty/clean comparisons where some comparisons between items maybe in correct, pointer predictions that allow us to index into the predecessor of an element in a data structure, and rank predictions where the rank of an element among a set of elements is noisily given.

For each prediction model, the authors improve upon classical algorithms which implement priority queues. For example with pointer predictions, one can insert elements in time proportional to the log of the error, compared to the classical $O(\log n)$ time (using skip lists). A similar result is shown for rank predictions, where a clever idea of reducing rank predictions to pointer predictions using an auxiliary vEB tree. Results for other settings such as sorting using predictions is also given as corollaries of their priority queue data structures, which matches and extends prior work.

The authors complement their upper bound with lower bounds, showing that for instance that in their dirty/clean comparison model, they obtain the optimal number of clean comparisons used for the extract min operation.

**Strengths:**

- The problem is quite motivated: it is known that in the worst case we cannot have all desired operations of a priority queue fast. So beyond worst case models are certainly relevant.
- The authors study three very natural models of predictions and give sounds justifications of their predictions.
- The randomized pivot idea in Theorem 2.1 is nice
- Section 3.3 makes a nice connection about using rank predictions to reduce the problem to integer keys (or bounded universe size), for which there often are better algorithms for. This idea could potentially have other application. It also gives some indication that rank queries maybe more powerful than pointer queries, since the authors use ranks to simulate pointers.
- The authors improve upon or match results for sorting with predictions studied in previous works.
- The experiments demonstrate that the, with appropriate predictions, the algorithms outperform worst case behavior

**Weaknesses:**

- The 'our results' section is a bit disorganized. It would be nice to have a clear table for the different prediction models, showing the prior worst-case bounds and then the bounds for the learning-augmented algorithms, for each of the operations. This would make it easier to quickly understand the improvements in each case, especially since different underlying data structures are used for different prediction models, and the number of results is large.
- There is a bit of context switching in section 3 since every subsection deals with a new prediction model. Maybe it would be better to not focus on binary heaps in section 2 as much, since the skip lists seem to be the main focus anyways.

**Questions:**

What does 'randomly filled positions in the leaf level' mean in algorithm 1? Does it mean the leaf elements are randomly permuted?

**Limitations:**

No societal impact.

---

> ### Author Rebuttal · Authors · 2024-08-05
>
> We thank the reviewer for their positive feedback and insightful suggestions. Below, we address the concerns and questions raised in the review.
>
> ### Weaknesses
> * **Table of results.** We will include a table, as suggested by the reviewer, summarizing the complexities of the operations using standard and learning-augmented priority queues. This would indeed make it easier to understand the improvements.
> * **Transitions in Section 3.** If the paper is accepted, an additional page will be allowed in the main body. This will provide us with enough space to create smoother transitions between the subsections that address the different models.
>
> ### Question
> * In a classical heap, the leaf level is filled from left to right. In our algorithm, whenever a new depth level is reached, denoting by $k$ the number of positions it contains ($k=n+1$), a uniformly random permutation $\sigma$ of $[k]$ is chosen. The leaf positions are then filled according to the order specified by $\sigma$. This is equivalent to choosing uniformly at random an empty position for each insertion operation.

---

> > ### Comment · Reviewer_unQV · 2024-08-09
> > **Response to authors**
> >
> > Thank you for the response. I would recommend to add the provided details about filling the heap in the final version. I maintain my score.

---

### Official Review · Reviewer_7XCi · 2024-07-12

**Soundness:** 3
**Presentation:** 3
**Contribution:** 2
**Rating:** 5
**Confidence:** 4

**Summary:**

The authors in this paper propose a learning-augmented priority queue data structure which takes advantage of the inaccurate predictions to facilitate the operations. Three different prediction models including dirty comparisons, pointer predictions, and rank predictions have been explored and discussed. The authors provide theoretical guarantees as well as empirical analysis in the paper, showing the superiority of their method.

**Strengths:**

1) The authors propose an interesting learning-augmented data structure by taking advantage of the predictions even when the predictions are not accurate. For the implementation wise, skip listed is used to overcome the inefficiency of the sorted linked list.

2) In addition, the authors have explored three different predictions and provide theoretical analysis to each one.

3) The authors have also discussed the potential applications of the proposed learning-augmented data structure and compared the performance with the SOTA method.

4) The authors have conducted numerical experiments and compared the performance with the SOTA method to show the superiority of the proposed learning-augmented priority queue.

**Weaknesses:**

1) For the comparison, the authors have compared their proposed priority queue with two traditional heaps. I found there is an advanced learning-augmented data structure in ref1, I am wondering how the proposed method would compare with that learning-based data structure.

2) Is there a threshold of the "inaccurate" prediction? For example, if after some threshold, will this proposed data structure be crashed?

3) It would be great if the authors could give more details about the experiments such as what is the scale of the experiments and how the model is set up.

**Questions:**

I have listed my concerns and questions in the [Strengths And Weaknesses] section.

**Limitations:**

I have listed my concerns and questions in the [Strengths And Weaknesses] section.

---

> ### Author Rebuttal · Authors · 2024-08-05
>
> We express our gratitude to the reviewer for their feedback and insightful comments. We address below the weaknesses they have raised.
>
> ### Weaknesses
> * **Comparison with prior learning-augmented algorithms.** We are uncertain about the specific data structure the reviewer is referring to and would appreciate a more precise reference. In any case, our work is the first to implement learning-augmented priority queues, making it difficult to compare directly with other data structures unless they support the same operations. However, we have compared the performance of our data structure for sorting, which is one possible application among many others, with previous learning-augmented sorting algorithms (Double-Hoover and Displacement Sort) as detailed in Sections 4 and 5.
> * **"Threshold of inaccurate predictions".** Could the reviewer give more clarifications on this question, in particular what is meant by "crashed"? Note that $\log \eta$ is at most $\log n$, hence the complexity of our learning-augmented priority queue is always at most that of a standard priority queue not using predictions, up to a constant factor.
> * **Details about the experiments.** For sorting, Figure 2 is obtained with $n=10^5$ as mentioned in the corresponding caption. For Dijkstra's algorithm, the number of nodes $n$ and the number of edges $m$ are indicated on each figure. More details about the setup, as well as additional experiments with different scales for both sorting and Dijkstra's algorithm can be found in Appendix G. If the paper is accepted, we will use the additional page to expand the experiments section with supplementary material from the appendix.

---

### Official Review · Reviewer_ZmGR · 2024-07-12

**Soundness:** 4
**Presentation:** 4
**Contribution:** 3
**Rating:** 7
**Confidence:** 5

**Summary:**

The paper considers designing data structures for priority queues that accept predictions / advice to improve the time complexity of common queue operations. The paper considers three different models of predictions - (i) dirty comparisons : cheap but possibly inaccurate comparisons are available. Goal is to utilize these cheap dirty predictions to reduce the reliance on expensive true comparisons; (ii) pointer predictions : predicted position of the predecessor of a key in the queue. (iii) rank predictions : predicted rank of the key in the universe of all keys.

The paper considers two priority queue implementations. (i) A binary heap is a simple structure that supports all priority queue operations in time O(log n). In particular, since inserting a new element involves inserting it at the correct position in a root-to-leaf path of length log n, it can be done via O(log log n) comparisons (but O(log n) time). The authors show that by using dirty comparisons instead insertion can be performed in O(log n) time using O(log log n) dirty comparisons and O(log log \eta) clean comparisons (in expectation) where \eta denotes the prediction error. The algorithmic ideas here are rather standard and unsurprising.
(ii) The second implementation considers skip lists - a probabilistic data structure that supports insertion in expected O(log n) time (and ExtractMin in O(1) time). In presence of predictions, the authors provide a new insertion algorithm that reduces the expected insertion time ( or #clean comparisons) to O(log \eta) where \eta is the corresponding error measure. Similar results are obtained for rank predictions.

Finally, the paper shows almost tight lower bounds in all models.

**Strengths:**

- The algorithms are very clean and easy to follow. The paper is well-written and the main ideas are readily accessible.
- Priority queues are a fundamental data structure and improving their performance via learning augmentation is a good contribution - e.g. prior results on learning augmented sorting are now simple applications.

**Weaknesses:**

Algorithmic techniques introduced are rather unsurprising and are mostly a collection of well known ideas

**Questions:**

N/A

**Limitations:**

Yes

---

> ### Author Rebuttal · Authors · 2024-08-05
>
> We thank the reviewer for their positive feedback and for the time and effort spent on our submission.
>
> **Weakness.**
> The intuitions behind some algorithmic ideas are indeed inspired by previous work, and we highlighted these connections as much as possible to make the paper and algorithms easier to understand. Given the efficiency of our algorithms, we believe that their simplicity can also be viewed as a strength rather than a weakness, although we understand the reviewer's perspective as well. However, the paper introduces several novel ideas. For instance, we demonstrate how to effectively use a van Emde Boas (vEB) tree to reduce the problem with rank predictions to the design of a priority queue within a finite universe, and how to handle the prediction errors in the subsequent analysis adequately.

---

> > ### Comment · Reviewer_ZmGR · 2024-08-09
> >
> > Thanks for the response! I maintain my positive score.

---

### Decision · Program_Chairs · 2024-09-25

**Decision:**

Accept (poster)

**Comment:**

The paper presents mechanisms for improving the known worse-case performance of priority queues via augmentation through learning.  The learning mechanisms presented are not new but putting them together to show performance improvement beyond the theoretical bound for three classes of problem presented is novel and is expected to have positive impact on the research community.